# C$_{16}$-ceramide is a natural regulatory ligand of p53 in cellular stress response

Baharan Fekry[1], Kristen A. Jeffries[1], Amin Esmaeilniakooshkghazi[1], Zdzislaw M. Szulc[2,3], Kevin J. Knagge[4], David R. Kirchner[4], David A. Horita [1], Sergey A. Krupenko[1,5] & Natalia I. Krupenko [1,5]

Ceramides are important participants of signal transduction, regulating fundamental cellular processes. Here we report the mechanism for activation of p53 tumor suppressor by C$_{16}$-ceramide. C$_{16}$-ceramide tightly binds within the p53 DNA-binding domain ($K_d \sim 60$ nM), in close vicinity to the Box V motif. This interaction is highly selective toward the ceramide acyl chain length with its C10 atom being proximal to Ser240 and Ser241. Ceramide binding stabilizes p53 and disrupts its complex with E3 ligase MDM2 leading to the p53 accumulation, nuclear translocation and activation of the downstream targets. This mechanism of p53 activation is fundamentally different from the canonical p53 regulation through protein–protein interactions or posttranslational modifications. The discovered mechanism is triggered by serum or folate deprivation implicating it in the cellular response to nutrient/metabolic stress. Our study establishes C$_{16}$-ceramide as a natural small molecule activating p53 through the direct binding.

[1] University of North Carolina at Chapel Hill Nutrition Research Institute, Kannapolis, NC 28081, USA. [2] Department of Biochemistry and Molecular Biology, Medical University of South Carolina, Charleston, SC 29425, USA. [3] Lipidomics Facility, Medical University of South Carolina, 173 Ashley Avenue, Charleston, SC 29425, USA. [4] David Murdock Research Institute, Kannapolis, NC 28081, USA. [5] Department of Nutrition, UNC Chapel Hill, Chapel Hill, NC 27599, USA. These authors contributed equally: Baharan Fekry, Kristen A. Jeffries. Correspondence and requests for materials should be addressed to N.I.K. (email: natalia_krupenko@unc.edu)

The p53 tumor suppressor is a key regulator of cell cycle, apoptosis, and survival in response to diverse stress stimuli[1]. It is a short-lived transcription factor with low basal levels in non-stressed cells maintained by constitutive proteasomal degradation[2]. The E3-ligase mouse double minute 2 (MDM2) plays the central role in this process pre-priming p53 for the degradation by ubiquitination at several lysine residues[2,3]. In cancer cells, the release of p53 from the complex with MDM2, controlled through protein–protein interactions and post-translational modifications, leads to the p53 accumulation and activation of downstream death signaling pathways[4–6]. Therefore, the p53–MDM2 interaction is an attractive target for chemotherapy, with a growing number of synthetic small molecule inhibitors of this interaction identified and tested as anticancer drugs[7,8]. Curiously, naturally occurring physiological small molecules disrupting the p53–MDM2 complex have not been reported, and the p53 activation in response to many drugs, nutrients, or metabolite changes is thought to be indirect and involve upstream protein regulators[9].

Bioeffectors interacting with p53 signaling pathways include ceramides, a group of sphingolipids consisting of >200 structurally distinct molecules[10,11]. Ceramides have emerged as important participants of signaling pathways regulating fundamental cellular processes, including proliferation, differentiation, death, and survival[12,13]. Their relationship with p53 pathways, though, remains elusive, with studies supporting both the function of ceramide upstream and downstream of p53[14]. We have recently reported that, in cancer cells, a transient elevation of one of the ceramide-generating enzymes, ceramide synthase 6 (CerS6),

upregulates p53 through an obscure mechanism[11]. Here we demonstrate the direct and highly specific interaction between p53 and $C_{16}$-ceramide, as well as a $C_{16}$-ceramide-based drug, both in vitro and in vivo. This interaction causes the loss of the p53–MDM2 complex and associated p53 ubiquitination, leading to the accumulation of p53. We also show that this type of the p53 regulation is engaged as the cellular response to serum starvation and as the adaptation to folate stress, findings implicating this mechanism in sensing nutrient stress and underscoring its physiological relevance.

## Results

**Soluble derivative of $C_{16}$-ceramide upregulates p53.** In our recent study, transient expression of CerS6 in cultured human cells increased $C_{16}$-ceramide and strongly elevated p53 protein[11]. To test whether ceramide itself has the same effect on p53, we treated several cancer cell lines with a water-soluble derivative of $C_{16}$-ceramide, $C_{16}$-pyridinium ceramide ($PC_{16}$, Fig. 1a). This structural analog of natural $C_{16}$-ceramide was shown to readily penetrate cellular membrane and accumulate in the cells[15]. In our experiments, $PC_{16}$ evoked strong dose-dependent elevation of p53, and its translocation to the nuclei (Fig. 1b, c). The p53 downstream targets p21 and PUMA were significantly elevated in response to $PC_{16}$ (Fig. 1b), further demonstrating the p53 activation.

**p53 binds $C_{16}$-ceramide with high affinity and selectivity.** We hypothesized that the above effect of $PC_{16}$ could be due to the direct interaction of this ceramide with p53. To test this

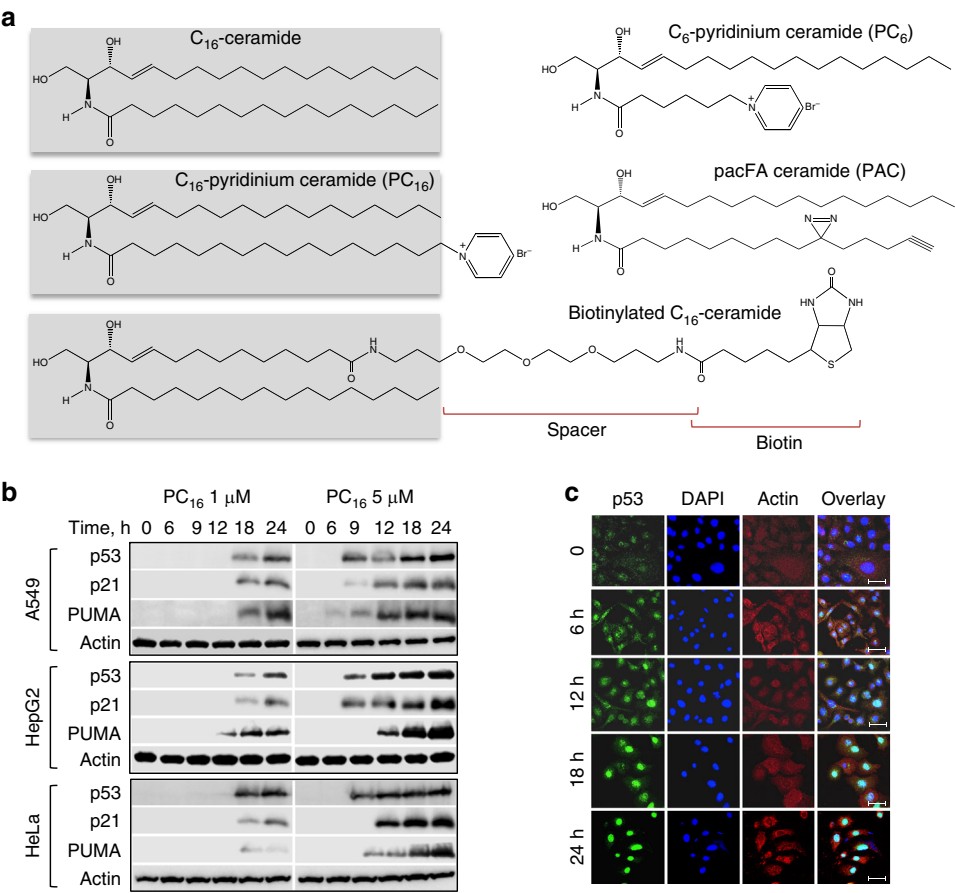

**Fig. 1** $C_{16}$-pyridinium ceramide activates p53. **a** Structures of ceramide compounds used in this study (the $C_{16}$-ceramide component is shown in gray box). **b** Treatment of cancer cells with $C_{16}$-pyridinium ceramide elevates p53 and its downstream targets in a concentration-dependent manner (western blot analysis). **c** p53 rapidly accumulates in the nuclei of A549 cells upon $C_{16}$-pyridinium ceramide treatment (confocal images, scale bar represents 40 μm)

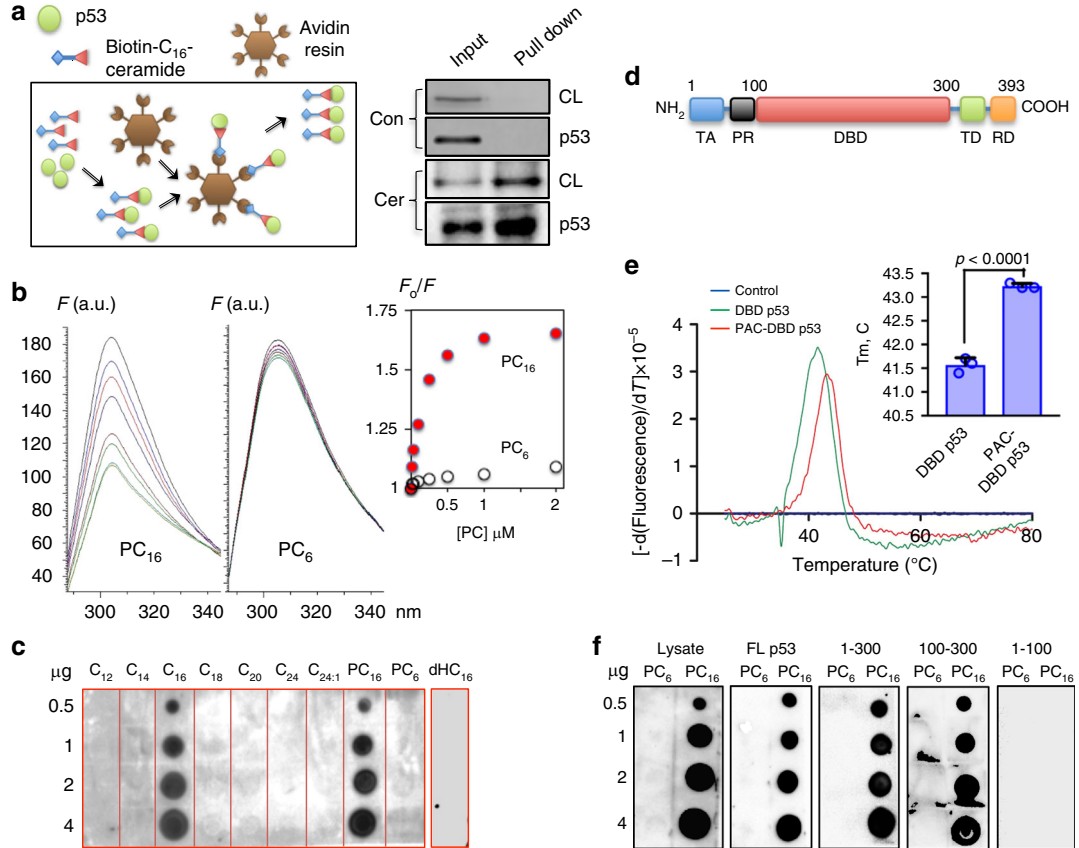

**Fig. 2** p53 binds $C_{16}$-ceramide within the core domain. **a** Left panel, schematic depicting p53 pull down on biotinylated ceramide. Right panel, pulled down preparations (from A549 cells lysate, CL, or pure recombinant p53) analyzed by SDS-PAGE/western blot with p53-specific antibody. For ceramide pull down (Cer), purified p53 or A549 cell lysates were incubated with biotinylated ceramide, while in control experiments (Con), both p53 and cell lysates were incubated with biotin. **b** Titration of p53 DBD (DNA-binding domain, the construct containing amino acid residues 100–300) tyrosine fluorescence with $PC_{16}$ (left panel) or $PC_6$ (middle panel). Right panel, data plotted as a function of the ceramide concentration. $F_O$ and $F$ are intrinsic p53 fluorescence in the absence or presence of ceramide, respectively. **c** Binding of purified p53 to a panel of ceramides with different acyl chain length ($C_{12}$–$C_{24}$ natural ceramides, $dhC_{16}$ natural $C_{16}$- dihydroceramide, $PC_6$ and $PC_{16}$ pyridinium ceramides with $C_6$ and $C_{16}$ acyl chains, respectively) immobilized on a PVDF membrane. Amounts of compounds in respective spots are indicated; bound p53 was detected with specific antibody. **d** Domain organization of p53 (TA transactivation domain, PR proline-rich domain, DBD DNA-binding domain, TD tetramerization domain, RD regulatory domain, numbers indicate amino acid residues). **e** Thermal shift assays of PAC-modified and unmodified p53 100–300. $T_m$ of unmodified p53 is 41.6 °C, $T_m$ of ceramide-modified p53 is 43.2 °C. Bar graph shows results of three independent experiments (each performed in triplicate) analyzed using Student's $t$ test. **f** Membrane-binding assays of p53 domains for the interaction with $PC_{16}$ (bound p53 was detected with specific antibody); numbers on the top indicate amino acid residues at the start and the end of truncated p53 constructs (Supplementary Figure 1); values on the left indicate amounts of ceramide spotted on the PVDF membrane. $PC_6$ was used as a negative control

hypothesis, we employed biotinylated derivative of $C_{16}$-ceramide (Fig. 1a), which in combination with avidin resin allows highly specific pull down of proteins interacting with $C_{16}$-ceramide (Fig. 2a). CaptAvidin Agarose used in these experiments efficiently binds biotin at pH 4.0 and releases it at pH 10.0, resulting in elution of biotinylated ceramide as well as ceramide-bound proteins, under very mild conditions. Using this affinity bait, we have pulled down endogenous cellular p53 from A549 cell lysates (Fig. 2a, right panel, Cer). To assure that the pull down of p53 was not the result of secondary interactions with other cellular proteins, which might interact with ceramide, we repeated the pull down using the purified recombinant full-length p53 protein (Supplementary Figure 1). In this experiment, p53 was pulled down as well (Fig. 2a, lower panel), proving that the interaction with ceramide bait was direct and not mediated by other proteins. In control experiments, purified p53 protein or cell lysate were incubated with non-modified biotin before loading to CaptAvidin Agarose. No p53 was pulled down in these experiments (Fig. 2a,

right panel, Con) indicating specificity of the pull down with the ceramide derivative of biotin.

The fluorescence titration of purified recombinant p53 with increased concentrations of $PC_{16}$ indicated a high affinity of this interaction, with the binding constant of about $60 \pm 20$ nM (Fig. 2b). Remarkably, another soluble ceramide derivative, $C_6$-pyridinium ceramide ($PC_6$, Fig. 1a), did not quench significantly the p53 fluorescence, indicating the lack of the high affinity interaction with the protein (Fig. 2b). This derivative has a significantly shorter acyl chain moiety compared to $PC_{16}$ (Fig. 1a), a structural feature most likely responsible for the loss of binding to the protein. The dramatic difference in the binding of two ceramide molecules also indicates that the cation–π stacking interactions (caused by the presence of pyridinium ring) within the p53 core domain[16] do not contribute to the high affinity complex formation. Overall, these experiments showed high selectivity of the p53 interaction toward the acyl chain length in the ceramide molecule.

The hydrophobic nature and low solubility of natural ceramide compounds in the aqueous media represent a challenge for characterization of their interactions with proteins in solution. Therefore, the selectivity toward $C_{16}$-ceramide has been further investigated using membrane-binding assays with purified p53. In this approach, 0.5–4 μg ceramide with different acyl chain lengths, from $C_{12}$ to $C_{24}$, were loaded on the polyvinylidene difluoride (PVDF) membrane, followed by blocking and incubation with the purified p53 protein (the purity of p53 preparations is shown in Supplementary Figure 1). After extensive washing, detection of p53 bound to membrane-attached ceramides was performed with a p53-specific antibody. Intriguingly, these experiments demonstrated the strong preference of p53 for $C_{16}$-ceramide: even minimal changes of acyl chain length completely abrogated the ability of p53 to interact with ceramide immobilized on the membrane (Fig. 2c). The only other ceramide capable of binding to p53, besides $C_{16}$-ceramide, was its soluble synthetic derivative, $PC_{16}$ (Fig. 2c; structures of the two ceramide molecules are shown in Fig. 1a). Another soluble derivative with $C_{18}$-acyl chain and pyridinium ring on the sphingoid base (previously shown to induce autophagy in cancer cells)[17] was unable to bind to p53 (Supplementary Figure 2). In agreement with fluorescence titration experiments, p53 did not interact with the pyridinium derivative having shorter acyl chain, $PC_6$, in the membrane-based assay (Fig. 2c). Importantly, $C_{16}$-dihydroceramide was also unable to bind to p53 (Fig. 2c) underscoring the importance of the sphingoid base conformation for the interaction with the protein.

**$C_{16}$-ceramide interacts with p53 within its core domain**. The p53 protein has five functional domains[18] (Fig. 2d) with the central DNA-binding domain (DBD) comprising the core of the protein[19]. Structural analysis of p53 indicated that 5+4 anti-parallel β-sheets form a β-sandwich thus creating a large hydrophobic central core of p53[20]. This type of β-sandwich structure has been characterized as a ceramide-binding motif[21]. In support of the binding of ceramide within the core domain, purified truncated p53 proteins (Supplementary Figure 1) lacking either the oligomerization domain (the construct including amino acid (aa) residues 1–300) or both the oligomerization and transactivation domains (the construct including residues 100–300, which constitute the DBD), bound $PC_{16}$ in membrane-binding assays similarly to the full-length protein (Fig. 2f). In agreement with this finding, the extended amino terminal transactivation domain itself (the construct including residues 1–100) did not bind ceramide (Fig. 2f).

To map the binding site of ceramide within the central DBD of p53, we utilized a photoactivatable ceramide (Fig. 1a, pacFA ceramide (PAC)). Upon binding of PAC to the protein, a diazirine group on the ceramide acyl chain enables the covalent modification of adjacent aa residues with ceramide, when exposed to the ultraviolet (UV) light. In our study, the covalent modification of the p53 DBD with this ceramide was confirmed by matrix-assisted laser desorption/ionization time-of-flight mass spectrometry (MALDI-TOF-MS; Fig. 3a), which detected a 528.3 Da increase in the molecular mass of the PAC-modified protein compared to the similarly treated protein in the absence of PAC. This increase was close to the expected theoretical mass difference. The MS-proteomics approach was further used to identify the site for ceramide modification in the p53 molecule (Fig. 3b). PAC-modified and non-modified p53 samples were denatured, reduced, alkylated, and digested with endoproteinase AspN. Protein digests were further subjected to liquid chromatography-tandem mass spectrometry (LC-MS/MS). PEAKS software analysis of the LC-MS/MS data revealed two

ceramide-modified residues, Ser240 and Ser241 (Fig. 3b), in the PAC-modified sample only. The AScore reflecting the probability for identified PAC modifications was 1000 (the highest possible). It should be noted that, because both residues, Ser240 and Ser241, were in the same peptide and because the fragmentation patterns were identical, this experiment did not differentiate between the two serine residues with regard to the ceramide modification. No ceramide-modified residues were identified in the control p53 sample digest (no ceramide added, Supplementary Figure 3). Of note, ceramide-modified and non-modified p53 samples showed differences in the peptide pattern after digestion, indicative of the altered accessibility of peptide bonds as a result of modification. To better visualize the ceramide-bound conformation, we docked $C_{16}$-ceramide to the p53 core domain using Autodock Vina. Figure 3c, left panel, shows a representative conformation of ceramide, modeled based on our experimental data, in which photoactivatable carbon (C10 of the ceramide acyl chain, shown in magenta) is proximal to Ser240 and Ser241. Importantly, in this model the bound ceramide is in the center of the second p53–MDM2 interface, which includes Box V motif of the core domain[22] (Fig. 3d). To prove that the identified residues are in immediate proximity to the bound ceramide, we generated series of p53 mutants where Ser240 or Ser241 were replaced with Ala, Lys, or Glu and tested these mutants for the ability to bind $PC_{16}$ (Supplementary Figure 4). The substitution of serine, a residue having a compact side chain, with more bulky side chain residues (Lys and Glu in the case of Ser 241 and Lys in the case of Ser 240) leads to the loss of ceramide binding (Supplementary Figure 4). At the same time, substitutions with aa having shorter side chains (S241A or S240E) did not affect the binding. These experiments confirm that aa residues 240 and 241 are involved in the accommodation of ceramide molecule in complex with p53. Additional residues are expected to be involved in the ceramide binding directly, through participation in the p53–ceramide interface, or indirectly by maintaining specific protein conformations. In this regard, the important question is whether residues in the DBD, commonly mutated in cancer, could be a part of the ceramide-binding network. While the complete answer to this question awaits further investigation, at least one of the most common mutations in the p53 DBD, R175H, did not affect the binding of $PC_{16}$ to p53 (Supplementary Figure 4a).

Nuclear magnetic resonance studies of the solution structure of the p53 core domain demonstrated that it is inherently unstable and has alternative hydrogen bond patterns in the protein interior[19]. Such arrangement allows for protein-fitting upon interaction with the binding partners such as DNA and protein co-activators. Thus dynamic stabilization/destabilization of p53 provides a regulatory mechanism for the protein function. For example, destabilization of the p53 structure and associated protein aggregation have been implicated in tumor induction/promotion[16]. In agreement with its intrinsic instability, the p53 protein melted just slightly above the body temperature in our experiments ($T_m$ of 41.6 °C for the DBD). Thermal shift assays further demonstrated the increase in $T_m$ of 1.66 °C for the DBD modified with ceramide ($T_m$ of 43.2 °C, Fig. 2e). These data not only support the binding of ceramide to the p53 DBD but also indicate a potential mechanism for the protein activation upon ceramide binding, which involves the protein stabilization.

**Ceramide disrupts p53–MDM2 complex preventing ubiquitination**. To address the biological relevance of the ceramide binding by p53, in view of proximity of the ceramide-binding site to the secondary p53–MDM2 contact region at the Box V motif, we examined ceramide effect on the p53–MDM2 interaction by

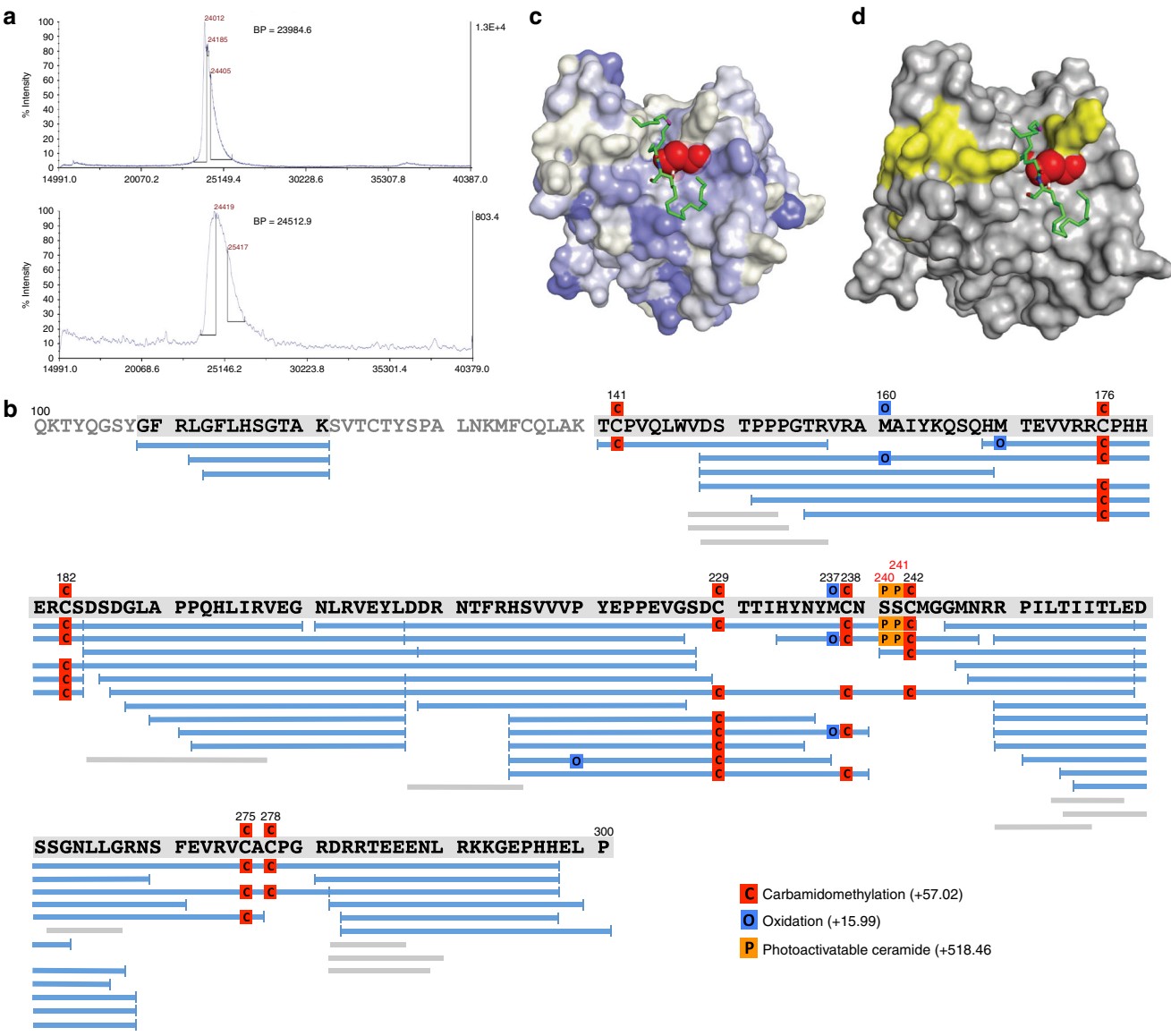

**Fig. 3** Binding site of ceramide within the p53 core domain. **a** MALDI-TOF MS of the control (non-modified) p53 construct containing residues 100–300 (top panel) and the same construct after PAC modification (bottom panel). Base peaks (BP) differ by 528.3 $m/z$. **b** PEAKS analysis of ceramide-modified p53 100–300. Ser-240 and Ser-241 were identified as modified (labeled with orange "P", AScore 1000). **c** Representative conformation of photoactivatable ceramide bound to the p53 construct (pdb 2mej). Hydrophobic surfaces are colored in blue, hydrophilic in white, and Ser240 and Ser241 in red. Photoactivatable ceramide is colored in green with the C10 atom shown in magenta. **d** Photoactivatable ceramide linked to Ser-240/-241 is in the center of the surface where p53 contacts MDM2[22] (shown in yellow). The p53 surface is colored in gray (pdb 2mej)

direct visualization of this complex in live cells using bimolecular fluorescence complementation approach. This approach allows generation of fluorescent signal from two non-fluorescent fragments of yellow fluorescent protein Venus[23] if they are in close proximity due to the interaction between p53 and MDM2 (schematically depicted in Fig. 4a). In our experiments, PC-3 cells co-transfected with V1-p53 and MDM2-V2 produced a clear fluorescent signal, which disappeared upon the treatment of cells with Nutlin 3, a well-known disruptor of the p53–MDM2 interaction (Fig. 4b and Supplementary Figures 5 & 6)[24]. PC-3 cells (p53[−/−]) were selected for these experiments because of the absence of endogenous p53 that should preclude the oligomerization of the V1-p53 monomers with the non-fluorescent endogenous p53. Similar to Nutlin 3, the treatment of cells with PC$_{16}$ led to the complete loss of fluorescence (Fig. 4b and

Supplementary Figures 5 & 6), indicating the lack of interaction between p53 and MDM2. In these experiments, drugs were applied for a short period of time (no more than 18 h) to avoid antiproliferative effects. Indeed, overlays of the phase-contrast images (Supplementary Figures. 5 & 6) shows that the number of cells and their density/morphology are the same in control and drug-treated samples, indicating that the loss of fluorescence upon treatment is not caused by cell death.

Complex formation between p53 and MDM2 in the absence or presence of either Nutlin 3 or PC$_{16}$ was also evaluated using purified proteins. In vitro measurements of the p53–MDM2 complex formation by sandwich enzyme-linked immunosorbent assay (ELISA; Supplementary Figure 7) directly demonstrated the loss of the interaction between p53 and MDM2 in the presence of PC$_{16}$, an effect similar to that observed for Nutlin 3

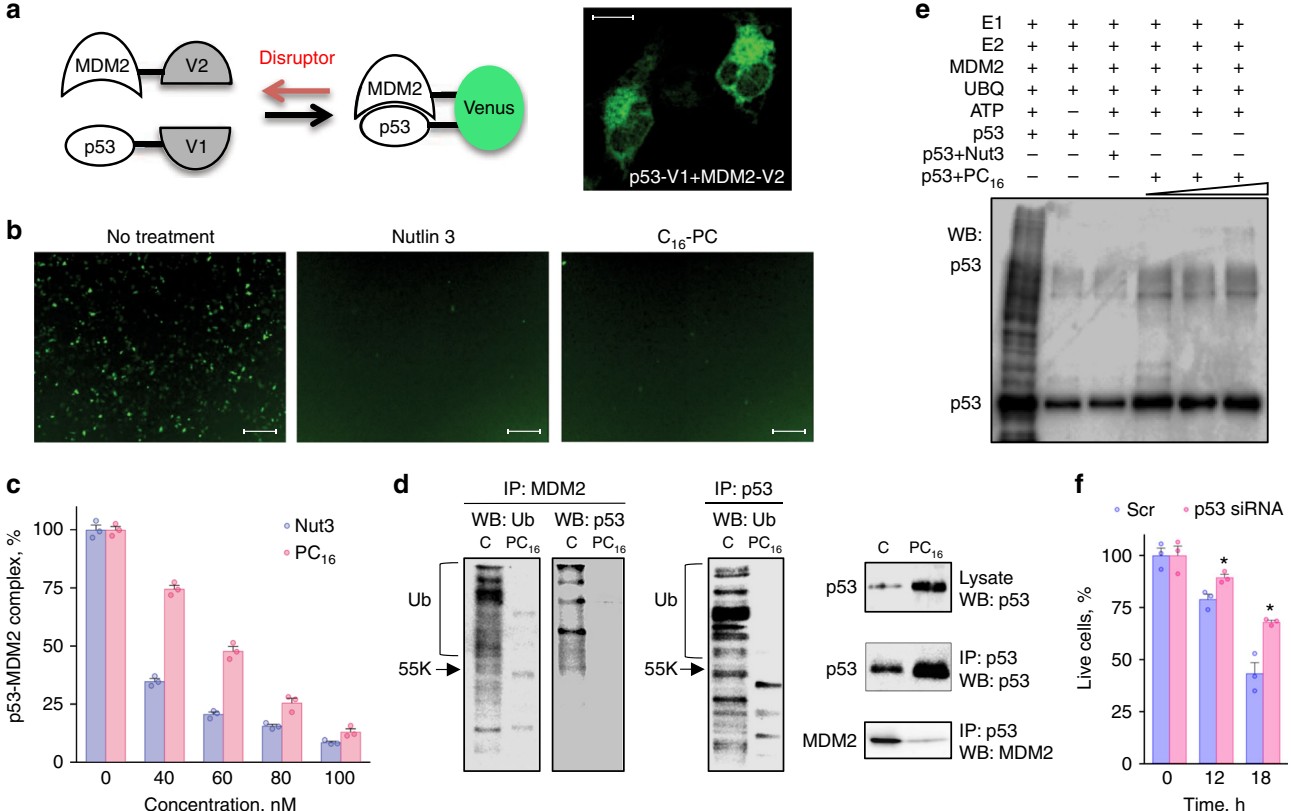

**Fig. 4** $C_{16}$-pyridinium ceramide is a disruptor of the p53–MDM2 complex. **a** Schematic of the approach to visualize p53–MDM2 complex in the cell; each of the proteins is expressed as a fusion with a half of yellow fluorescence protein Venus (V1 and V2, correspondingly); the formation of the complex between p53 and MDM2 reconstitutes Venus (seen in the confocal image of PC-3 cells, scale bar represents 20 μm; see also Supplementary Figures 5 & 6). **b** Fluorescent microscopy of untreated and Nutlin 3 (50 μM) or $PC_{16}$ (5 μM) treated PC-3 cells co-expressing p53-V1 and MDM2-V2. Scale bar represents 200 μm. **c** ELISA measurements of the p53–MDM2 complex in the presence of increasing concentrations of either Nutlin 3 or $PC_{16}$ (mean ± SE, $n = 3$; standard curve is presented in Supplementary Figure 4). **d** p53 does not co-immunoprecipitate with MDM2 (left panel) and loses ubiquitination (middle panel) in $PC_{16}$-treated cells; right panel shows input. Reciprocally, MDM2 very weakly co-immunoprecipitates with p53 in $PC_{16}$-treated cells (right panels). A549 cell lysates were used in these experiments. **e** In vitro ubiquitination assays of purified p53 alone and in the presence of Nutlin 3 (80 nM) or $PC_{16}$ (60, 80, and 100 nM). **f** siRNA silencing of p53 partially rescues A549 cancer cells from the $PC_{16}$ toxicity. Error bars represent ± SE, $n = 3$ (*$P < 0.01$, Student's $t$ test). A representative experiment (out of three) is shown

(Fig. 4c). In agreement with this phenomenon, the lack of p53–MDM2 complex in A549 cells treated with $PC_{16}$, compared to untreated (control) A549 cells, was demonstrated in the reciprocal pull down assays. Levels of p53 pulled down with MDM2 from $PC_{16}$-treated compared to non-treated cells were essentially undetectable (Fig. 4d, left panels). Likewise, the p53 pull down demonstrated significantly lower levels of p53-bound MDM2 for treated cells, compared to controls (Fig. 4d, right panels). In line with this finding, $PC_{16}$-treated cells revealed a strong decrease in levels of ubiquitinated p53 (Fig. 4d, central panel).

Further evidence for the $PC_{16}$ effect on the p53 ubiquitination was obtained in the in vitro ubiquitination assays utilizing purified proteins. These experiments demonstrated that $PC_{16}$ was as efficient in preventing p53 ubiquitination as Nutlin 3 (Fig. 4e). The relevance of ceramide-dependent mechanism of p53 regulation has been also demonstrated in live cells: the small interfering RNA (siRNA) silencing of p53 prevented $PC_{16}$ cytotoxicity in A549, HCT116, and HepG2 cell lines having wild-type (WT) p53 (Fig. 4f and Supplementary Figure 8). Importantly, p53-deficient cells were also insensitive to the elevation of endogenous $C_{16}$-ceramide as well[11].

**Endogenous $C_{16}$-ceramide regulates p53.** While our experiments present evidence that p53 physically interacts with

exogenous ceramide, we next asked the question of whether endogenous cellular $C_{16}$-ceramide interacts with p53 and regulates its activity. In unstressed cells, endogenous $C_{16}$-ceramide levels are rather low but were significantly elevated upon CerS6 transient expression (Fig. 5a). We have generated lysates from A549 cells transiently expressing CerS6 and non-transfected (control) A549 cells and used one part of each lysate for the p53 pull down with p53-specific antibody (to evaluate specific ceramide binding to this protein), while the remaining lysate was used for the actin pull down with actin-specific antibody (to evaluate potential non-specific ceramide binding). Elution with glycine buffer, pH 3.0, was used to release pulled down proteins from Protein G agarose, a procedure that provides increased specificity of the pull down. Protein levels in the pulled down fractions were evaluated by western blotting (WB), and ceramide levels in the pulled down proteins were measured by LC-MS/MS. All ceramide measurements were normalized by the protein levels in the respective pull downs. The analysis of sphingolipid profiles of p53 immunoprecipitated from A549 cells demonstrated that in non-stressed cells a very low amount of $C_{16}$-ceramide was bound to p53 (Fig. 5b). However, in response to the CerS6 expression, which resulted in the elevation of intracellular $C_{16}$-ceramide (Fig. 5a), significant amounts of $C_{16}$-ceramide were detected in pulled down p53 (Fig. 5b), indicating that ceramide was bound to the protein in vivo. The analysis of the extended sphingolipid

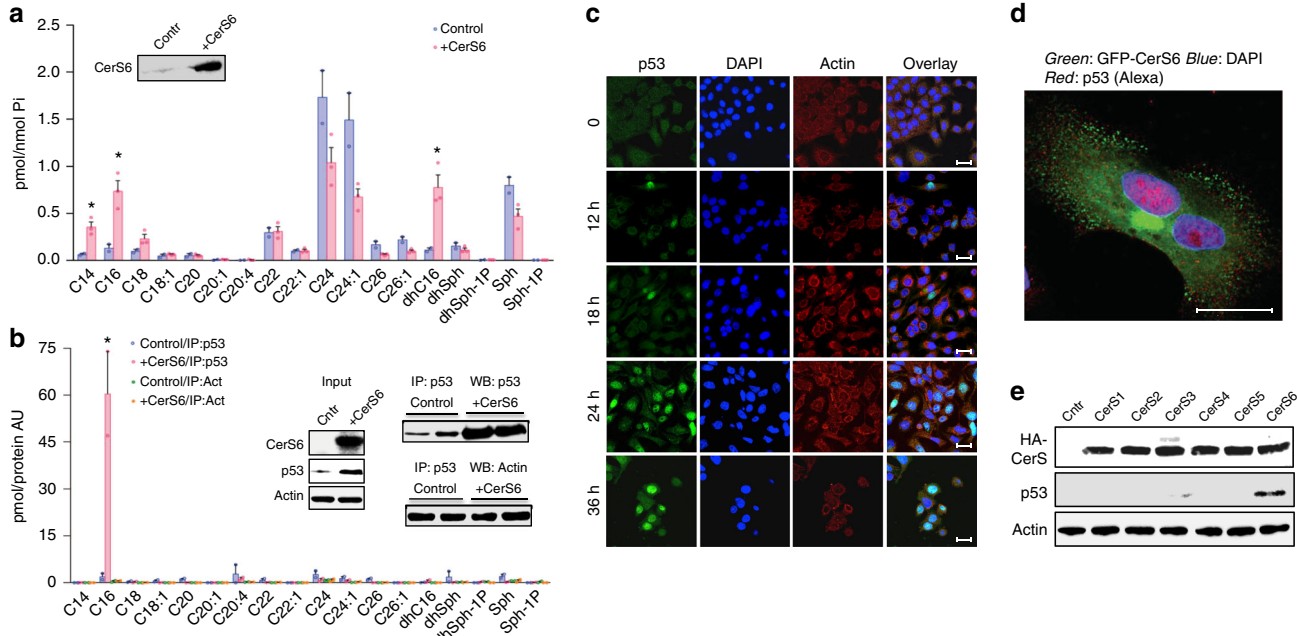

**Fig. 5** p53 forms complex with natural $C_{16}$-ceramide in the cell. **a** Levels of ceramide species in A549 cells after CerS6 transient transfection (36 h) or in non-transfected control cells (LC-MS/MS; mean ± SD; $n = 2$; *$P < 0.0002$, Student's $t$ test). **b** Ceramide levels (mean ± SD; $n = 2$; *$P < 0.01$, ANOVA) in p53 and actin fractions pulled down with corresponding antibodies from lysates of control or CerS6-transfected A549 cells. Lysates were divided into two portions, one was used for pull down with p53 specific antibody and the other used for pull down with actin-specific antibody (inset). Ceramide measurements were normalized to the amounts of specific proteins in the pull downs. **c** Transient expression of CerS6 in A549 cells induces rapid accumulation of p53 and its nuclear translocation (green, p53; red, actin/cytoplasm; blue, DAPI/nuclei; scale bar represents 20 μm). **d** Elevation of CerS6 in A549 cells (GFP-CerS6, green) leads to mostly nuclear localization of p53 (red); single cell image, scale bar represents 20 μm. **e** Expression of CerS6 but not CerS1-5 induces p53 elevation in A549 cells 48 h post-transfection

profiles has shown that $C_{16}$-ceramide was the only ceramide found in the complex with p53 in significant amounts (Fig. 5b). At the same time, levels of $C_{16}$-ceramide were not the highest in analyzed cells (levels of $C_{24}$ and $C_{24:1}$ were the same or higher) (Fig. 5a). The lack of other ceramide species bound to p53, as well as the absence of dh$C_{16}$-ceramide in the complex, confirm the specificity observed in the membrane binding assays. The interaction with $C_{16}$-ceramide was specific to p53: the preparation obtained by the pull down of actin from the same cells did not demonstrate either the enrichment with $C_{16}$-ceramide upon CerS6 expression or prevalence of this ceramide in the pull down compared to other ceramide species (Fig. 5b). In agreement with the CerS6-mediated $C_{16}$-ceramide accumulation, p53 translocated to nuclei in CerS6-expressing cells (Fig. 5c, d) indicating the protein activation. The family of ceramide synthases in humans consists of six members, CerS1–6, which synthesize ceramide molecules with different acyl chain length[25] (Supplementary Figure 9). Remarkably, only CerS6 was capable of activating p53 upon transient transfection, while the elevation of CerS1–5 in the same manner did not result in the p53 increase (Fig. 5e and Supplementary Figures 9 & 10d). Of note, ceramide measurements in the cells transfected with individual ceramide synthases show good correlations between elevated ceramide species and the specificity of the expressed ceramide synthase (Supplementary Figure 9). The only exception was CerS3, which synthesizes ultra-long-chain ceramides. CerS3, however, is mostly expressed in the skin and testes[26] where pathways specific to these tissues provide ultra-long-chain acyl-CoA substrates for the enzyme. If corresponding pathways are not active in cancer cells, this would explain the lack of the enzyme product in our experiments. As expected, $C_{16}$-ceramide was elevated in cells transfected with CerS5 plasmid, though ~2-fold lower than upon transfection with CerS6. However, no p53 activation was observed in either A549

or HCT116 cells in response to CerS5 expression (Fig. 5e and Supplementary Figures 9 & 10d).

**CerS6-dependent p53 activation upon serum starvation.** The physiological relevance of the CerS6/$C_{16}$-ceramide dependent mechanism of p53 regulation is further demonstrated in the study of cellular response to serum starvation. As expected, serum withdrawal causes the elevation of p53 and its downstream target p21 (Fig. 6a) leading to strong $G_0/G_1$ cell cycle arrest in A549 cells (Fig. 6b). Importantly, in serum-deprived cells CerS6 and $C_{16}$-ceramide were also elevated (Fig. 6a, c). The follow-up experiments demonstrated that the CerS6 elevation is a common response to serum withdrawal in several cell lines, with $C_{16}$-ceramide showing the most profound increase among the ceramide species (Fig. 6c and Supplementary Figures 10a & 11a). In agreement with previous findings, p53 pull downs from serum-starved cells (A549, HCT116, and HepG2 cell lines were used in these experiments) contained significantly higher amounts of $C_{16}$-ceramide than such pull downs from non-stressed cells (Fig. 6d and Supplementary Figures 10b & 11b) indicating that in stressed cells p53 is in complex with this ceramide. Strikingly, the siRNA silencing of CerS6 prevented both p53 and p21 accumulation in response to serum withdrawal and partially rescued cells from $G_0/G_1$ arrest (Fig. 6a, b and Supplementary Figures 10c & 11c). Accordingly, CerS6-silenced cells did not elevate $C_{16}$-ceramide upon serum withdrawal, and the p53 pulled down from these cells has much lower bound $C_{16}$-ceramide, similar to values in unstressed cells (Fig. 6c, d and Supplementary Figures 10a, 10b, 11a & 11b). These experiments clearly demonstrate that the p53-dependent response to serum starvation is mediated in part by CerS6-produced $C_{16}$-ceramide. Of note, our previous work implicated CerS6 and $C_{16}$-ceramide elevation in the cellular response to folate stress[11]. Similar to serum starvation experiments, p53 pulled down from cells experiencing folate stress

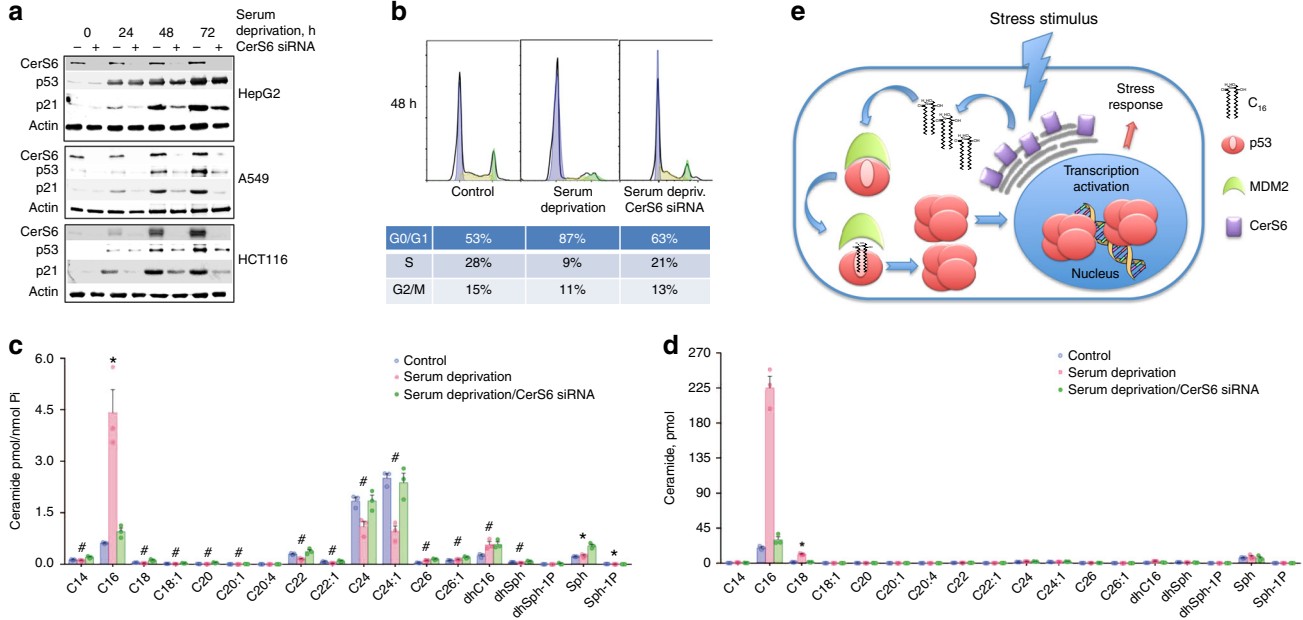

**Fig. 6** CerS6-dependent $C_{16}$-ceramide generation mediates stress response to serum starvation. **a** Serum withdrawal elevates CerS6 in cultured cells with concomitant increase of p53 and p21 while CerS6 siRNA silencing prevents the p53 and p21 accumulation in response to serum withdrawal. **b** CerS6 siRNA silencing partially rescues $G_0/G_1$ cell cycle arrest upon serum starvation. **c** Serum withdrawal elevates $C_{16}$-ceramide in A549 cells (48 h post-withdrawal); this effect was abrogated upon CerS6 silencing (mean ± SE; $n = 3$; *$P < 0.001$; #$P < 0.05$, ANOVA). **d** p53 is pulled down in complex with $C_{16}$-ceramide from serum-starved A549 cells but not from serum-starved cells with silenced (siRNA) CerS6 (mean ± SE; $n = 3$; *$P < 0.00001$, ANOVA). **e** Proposed mechanism of p53 regulation by $C_{16}$-ceramide: cellular stress elevates CerS6 in endoplasmic reticulum, leading to generation of $C_{16}$-ceramide ($C_{16}$); ceramide binds to p53, preventing its interaction with MDM2; this halts the p53 ubiquitination and degradation; resulting accumulation of p53 leads to its nuclear translocation and transcriptional activation of stress response proteins

(due to elevation of the folate-stress enzyme ALDH1L1[11,27]) had high levels of bound $C_{16}$-ceramide as well (Supplementary Figure 12).

## Discussion

Ceramides are increasingly recognized as participants in signal transduction but the exact mechanisms of their effects are still not completely understood. One of the possible mechanisms enabling the regulatory function of ceramides through the direct interaction with key components of signaling pathways was demonstrated for protein kinase C[28–30], kinase suppressor of Ras[31], LC3B-II[17], and the PP2A inhibitor I2PP2A[32]. This regulation was executed by different ceramide species, the finding suggesting strict specificity of the signaling effect of ceramide molecules depending on their respective protein targets. The connection between ceramide signaling and p53 has been known for over a decade but the underlying mechanisms remain obscure. While most effects of p53 on ceramide signaling are predominantly associated with the transcriptional regulation of sphingolipid pathway enzymes, the effects of ceramide on p53 are complex and not well understood, with different studies reporting conflicting results (reviewed in ref.[33]). For example, treatment of HCT116 cells with exogenous $C_{16}$-ceramide transcriptionally upregulated p53[34], while CerS5 expression in COS-7 cells repressed the p53 reporter activity[35]. Several studies demonstrated indirect effect of ceramide on p53, through additional mediators such as PP2A, AMPK, and Bcl-2[36,37], but the mechanistic studies of the direct ceramide effect on p53 are still missing.

Function of p53 in the cell is highly regulated, with the E3 ubiquitin ligase MDM2 playing a critical role in this process[2,3,38]. Numerous posttranslational modifications of p53 as well as several drugs can release the protein from the complex with MDM2 thus preventing its ubiquitination and proteasomal

degradation[4,7,39,40]. In this study, we present evidence that p53 tightly binds $C_{16}$-ceramide, an event disrupting the protein interaction with MDM2. The subsequent loss of the ubiquitination of p53 prevents its proteasomal degradation leading to protein accumulation and activation of the downstream targets. Together with canonical ways of p53 activation via posttranslational modifications and interaction with protein co-regulators, this mechanism further adds to the complexity as well as plasticity of the p53 signaling.

In general, p53 is not known for binding physiological metabolites. A notable exception is NAD$^+$, which apparently binds to p53 with a millimolar range affinity constant and attenuates the DNA binding[41]. Our finding is an example of a natural cellular metabolite binding to p53 to regulate its degradation. This effect is similar to the effect of nutlins, small synthetic molecules interacting with the p53-binding partner MDM2[7,24,40]. Another non-physiological small molecule identified in a high throughput screen, RITA, binds with high affinity within the transactivation domain of p53 and disrupts its interaction with MDM2[42] (though a recent study has indicated that RITA functions not as p53 activator but rather as genotoxic DNA cross-linking drug[43]). In contrast to RITA, cellular sphingolipid $C_{16}$-ceramide generated in response to cellular stress binds in the core domain of p53.

Our studies identified two aa residues in this domain, Ser240 and Ser241, to be in close proximity to the C10 atom of the ceramide acyl chain. This specific site is crucial for the ceramide accommodation since the residue substitutions introducing bulky side chains prevent the complex formation (Supplementary Figure 4). Modeling of the $C_{16}$-ceramide into the p53 DBD with the C10 atom of the ceramide acyl chain proximal to Ser240 or Ser241 demonstrates that this sphingolipid can be accommodated in the hydrophobic groove in the vicinity of these residues (Fig. 3c). Though the modeling approach still has limitations even with the narrowly defined position of the C10 atom of ceramide

acyl chain, curiously, our model places ceramide right in the middle of the secondary site forming the p53–MDM2 interface[44], in close proximity to the BOX V motif of p53 (Fig. 3d). Importantly, the BOX V motif was implicated in the MDM2 binding[22,44] as a part of the secondary site, which interacts with the central region of MDM2 responsible for the ubiquitination and degradation of p53[45,46]. Thus ceramide binding to p53 could stabilize the protein preventing the conformational adaptation that forms the interface for the MDM2 interaction. Alternatively, the placement of ceramide in the middle of p53–MDM2 binding region could create a physical barrier for the interaction.

Remarkably, our experiments show that p53 is highly selective for $C_{16}$-ceramide, an unexpected finding taking into consideration the overall structural similarity of ceramide molecules with different acyl chain length. It is difficult to explain the basis for such high specificity toward the acyl chain length without knowing the precise conformation of the bound ceramide molecule as well as the overall structure of the p53–ceramide complex. Curiously though, high selectivity for the acyl chain length has been also demonstrated for the binding of $C_{18}$-ceramide by the PP2A inhibitor I2PP2A, which was not capable of binding $C_{16}$-ceramide[32]. Thus the phenomenon of differentiation between acyl chains differing by two carbon atoms may not be a unique feature of the p53–ceramide binding but a more general mechanism of sphingolipid signaling. The inability of p53 to bind dihydro-$C_{16}$-ceramide further underscores very specific requirements for the ligand in terms of not only the acyl chain length but also of the chains conformation/flexibility.

Such high specificity indicates that p53-dependent pathways will not respond to the overall changes in ceramide or dihydroceramide levels but can be triggered upon changes of the specific ceramide species, $C_{16}$-ceramide. While the elevation of ceramide in general is viewed as the pro-apoptotic event[47], the mechanism discovered in this study indicates that $C_{16}$-ceramide could be an ultimate discriminator among these bio-effectors defining the cellular response to stress stimuli. Toward this end, CerS6, the key enzyme producing this ceramide, is typically expressed at low levels in normal tissues[25] but is upregulated in response to stress[11,48]. Interestingly, the elevation of another $C_{16}$-ceramide-producing enzyme, CerS5[25], did not activate p53. This phenomenon was observed in several cell lines (Fig. 5e and Supplementary Figures 9 & 10d) indicating a general lack of the p53 response to CerS5. Though the basis for this phenomenon is not clear, it could be explained by insufficient $C_{16}$-ceramide production from CerS5 to engage the activation (threshold effect). Indeed, the elevation of $C_{16}$-ceramide above the basal level was 2.6-fold in the case of CerS6 expression and only 1.75-fold in the case of CerS5 expression. Alternatively, differences in the precise localization and membrane orientation, the selection of interacting partners (in the membrane as well as in the cytosol), and the ability to release $C_{16}$-ceramide to a specific target could play a role in the CerS6-dependent but CerS5-independent mechanism of the p53 regulation. Overall, our findings place CerS6 in the position of a very specific mediator for certain types of stress responses.

The mechanism reported here is likely to have a more general implication. For example, in our study, HeLa cells having WT p53 responded to $PC_{16}$ treatment similarly to other cells with the WT p53 protein. However, in these cells the protein is regulated not by MDM2 but by HPV E6[49]. The regulation of p53 by the viral E6 protein is highly similar to the regulation via MDM2, with E6 recruiting a different E3 ligase, E6AP (UBE3A), to ubiquitinate p53 for subsequent proteasomal degradation[49]. Recently solved crystal structure of the trimeric complex of p53, E6, and E6AP showed that the E6 protein makes contacts with the p53 DBD via four residues, N268, E286, L289, and R290[50]. The analysis of the

structure of this complex indicates that these residues flank the Box V motif of p53, which also represents a part of the p53/MDM2 interface affected by the ceramide binding. We suggest that in HeLa cells the binding of ceramide to p53 prevents the interaction with E6–E6AP complex thus imposing protective effect from ubiquitination and proteasomal degradation in a fashion similar to the protection from MDM2. While this regulatory mechanism explains the induction of apoptosis by $C_{16}$-ceramide in $p53^{+/+}$ cancer cells, it could as well be pertinent to the regulation of homeostasis in normal non-cancer cells, which typically maintain low basal levels of p53. Upon metabolic stress or nutrient starvation, sensed by the sphingolipid pathways, elevated ceramide transmits the signal to p53 evoking such cellular responses as cell cycle arrest, apoptosis, autophagy, or senescence. In line with this mechanism, serum deprivation of cultured cells causes the elevation of $C_{16}$-ceramide[51], a phenomenon also demonstrated in our study.

It should be noted that the connection between the p53 regulation and the ceramide pathway is more complex and is not limited to the mechanism uncovered by the present study. Thus our previous work has demonstrated that CerS6 itself is a transcriptional target of p53[11,52]. Together with the CerS6 and $C_{16}$-ceramide-dependent mechanism of the p53 activation described here, such regulation creates a positive feedback loop, which has the capacity to robustly multiply stress signals to produce an efficient and swift response associated with elevation of p53. There are still several important yet unanswered questions regarding the phenomenon of the ceramide binding to p53, including: the mechanism of the ceramide loading on p53 in the cell; the effect of ceramide binding on the function of p53 mutants; the effect of p53 modifications on ceramide binding and vice versa; and the fate of the ceramide–p53 complex with regard to its sustainability, interactions with p53 protein partners, and the transactivation function of the protein. The answers to these questions will help to clarify the reciprocal regulation of p53 and ceramide pathways as well as the role of their cross-talk in maintenance of cellular homeostasis.

Overall, our study establishes the specific sphingolipid, $C_{16}$-ceramide, as a physiological small molecule ligand of p53 regulating the protein ubiquitination and proteasomal degradation. This mechanism also clarifies the role of CerS6 in the induction of apoptosis in a p53-dependent manner and links ceramide pathways to the p53-dependent response to nutrient deprivation in cancer as well as normal cells that might impact therapeutic approaches targeting human cancers.

## Methods

**Cell culture and reagents**. The HCT116 $p53^{-/-}$ cell line isogenic to HCT116 ($p53^{+/+}$, obtained from ATCC) was a generous gift from Dr. Vogelstein. Cell culture media and reagents were purchased from Invitrogen. A549 and PC3 cells (both from ATCC) were grown in RPMI 1640 supplemented with 2 mM glutamine, 1 mM sodium pyruvate, and 10% fetal bovine serum (FBS, Atlanta Biologicals). HepG2 and HeLa cells (obtained from ATCC) were propagated in Eagle's Minimum Essential Medium with 10% FBS. All cells were grown at 37 °C under humidified air containing 5% $CO_2$.

$C_{12}$-ceramide (N-(dodecanoyl)-sphing-4-enine), $C_{14}$-ceramide (N-(tetradecanoyl)-sphing-4-enine), $C_{16}$-ceramide (N-(hexadecanoyl)-sphing-4-enine), $C_{18}$-ceramide (N-(octadecanoyl)-sphing-4-enine), $C_{20}$-ceramide (N-(eicosanoyl)-sphing-4-enine), $C_{24}$-ceramide (N-(tetracosanoyl)-sphing-4-enine), $C_{24:1}$-ceramide (N-(15Z-tetracosenoyl)-sphing-4-enine), and photoactivatable ceramide (N-(9-(3-pent-4-ynyl-3-H-diazirin-3-yl)-nonanoyl)-D-*erythro*-sphingosine) were from Avanti Polar Lipids. $PC_6$ (D-*erythro*-2-N-[6′-(1″-pyridinium)-hexanoyl]-sphingosine bromide) and $PC_{16}$ (D-*erythro*-2-N-[16′-(1″-pyridinium)-hexadecanoyl]-sphingosine bromide) as well as $C_{16}$-biotinylated ceramide (D-*erythro*-2-N-[hexadecanoyl]-sphingosine-14′-(ω-biotin-NH-dPEG3-NHCO)) and $C_{18}$-ceramide with the pyridinium ring on sphingoid base ((2s,3R)-2-N-octadecanoyl-14-(1′-pyridinium)-sphingosine bromide) were synthesized by MUSC Lipidomics Shared Resource/Synthetic Core (http://www.hollingscancercenter.org/research/shared-resources/lipidomics/index.html). Nutlin 3 was from Santa Cruz.

**Transient transfection**. A549, HepG2, and PC-3 cells ($2 \times 10^6$) were transfected with 2 μg of corresponding expression vector using a Neon transfection system (Invitrogen) according to the manufacturer's protocol. In control experiments, respective "empty" plasmids were used for transfection. Vectors for the bimolecular fluorescent complementation pCS2-P53-V1, pCS2-P53-V2, pCS2-MDM2-V1, and pCS2-MDM2-V2 were a generous gift from Professor Cecilia Rodrigues. Vectors pcDNA3.1/p53 (WT p53 expression) and pCMV/CerS6 (CerS6 expression) were described previously[11]. Vectors pcDNA3.1/CerS2-HA, pcDNA3.1/CerS3-HA, pcDNA3.1/CerS4-HA, pcDNA3.1/CerS5-HA, and pCMV2b/FLAG-CerS1 were a generous gift from Professor Anthony Futerman.

**Microscopy**. Cells grown in Lab-Tek II Chamber (Nulg Nunc International) were fixed with 3.7% methanol-free formaldehyde for 10 min and permeabilized with 0.1% Triton X-100 for 5 min. After blocking with 10% pre-immune goat/chicken serum in phosphate-buffered saline (PBS) for 45 min, slides were incubated with p53-specific antibody FL393 (1:200, Santa Cruz Biotechnology, sc-6243) and actin monoclonal antibody (1:1000, Sigma a5441) overnight at 4 °C. Slides were washed with TBST buffer followed by incubation with secondary chicken anti-rabbit antibody conjugated with Alexa Fluor 488 (1:500, Life Technologies A-21441) and goat-anti-mouse IgG antibody conjugated with Alexa Fluor 555 (1:500, Life Technologies A32727) in a dark chamber for 45 min at room temperature. Nuclei were stained by 4,6-diamidino-2-phenylindole (Invitrogen) according to the manufacturer's protocol. Images were captured using laser scanning confocal microscope Carl Zeiss LSM 710 (Carl Zeiss) at the David H. Murdock Research Institute (DHMRI).

**Cell proliferation assay**. Viable cells were assessed using an 3-[4,5-dimethyl-thiazol-2-yl]-2,5 diphenyl tetrazolium bromide (MTT) cell proliferation assay (Promega). Cells were seeded in 96-well plates at a density of $5 \times 10^3$ cells/well and MTT was added at specific time points. Plates were further processed according to the manufacturer's instructions. $A_{570 \text{ nm}}$ was measured using a Wallace 1420 multilabel counter (PerkinElmer Life Sciences).

**Immunoblot analyses**. For WB analysis, cells were lysed in a 50 mM Tris-HCl buffer, pH 8.0 containing 150 mM NaCl, 2 mM EDTA, 1% Triton X-100, 0.1% sodium dodecyl sulfate (SDS), 1 mM dithiothreitol, 1 mM phenylmethanesulfonylfluoride, and protease inhibitor cocktail (Sigma). Cell lysates were subjected to SDS-polyacrylamide gel electrophoresis (PAGE) followed by immunoblot with corresponding antibodies. CerS6 polyclonal antibody (1:1000) was from Novus Biologicals, NBP1-76965. Monoclonal MDM2 (SMP14, 1:200, sc-965) and p53 (DO1, 1:400, sc-126) antibodies as well as polyclonal p53 (FL-393, 1:400, sc-6243) and p21 (M-19, 1:400, sc-471) antibodies were purchased from Santa Cruz Biotechnology. PUMA polyclonal antibody (1:1000, P4618) and monoclonal actin antibody (AC-15, 1:5000, a5441) were from Sigma. Full blot scans are provided in Supplementary Information, Full Blot Scans section.

**Biotinylated ceramide/CaptAvidin pull down**. Purified recombinant p53 protein or lysate from A549 cells (obtained from ATCC) transfected with pcDNA3.1/p53 plasmid were incubated with biotinylated $C_{16}$-ceramide (MUSC Lipidomics Shared Resource https://lipidomics.musc.edu/web/guest/lipidomics-core) overnight and mixed with CaptAvidin agarose (Life Technologies) in biotin-binding buffer (50 mM citrate phosphate, pH 4.0); the pull down was carried out according to the manufacturer's manual. The elution of the biotinylated ceramide was performed in mild conditions (50 mM sodium carbonate–HCl buffer, pH 10); p53 in eluted samples was detected by SDS-PAGE/immunoblot assays.

**Membrane-binding assays**. To evaluate binding of p53 to various ceramides, 10 mM stock solutions of natural ceramides in methanol or pyridinium ceramides in dimethyl sulfoxide (DMSO) were spotted on the PVDF membrane in the indicated amounts and dried. Membranes were blocked with 3% bovine serum albumin (BSA) in TBST buffer for 1 h at room temperature and then incubated overnight at 4 °C with either purified p53 protein or cell lysate from p53 transfected cells. Membrane was washed with TBST buffer at room temperature and the bound p53 was detected with p53-specific Ab (FL-393, 1:400). To detect binding of the truncated variants of p53, DO1 MAb (1:400) for the 1–100 aa construct and a monoclonal antibody PAb 240 (1:800, Invitrogen, 13-4100) for all other constructs were used. Odyssey Fc scanner (LI-COR Biosciences) was used for visualization of bands/spots after immunodetection and image analysis was performed with the Image Studio software v.3.1 (LI-COR).

**Antibody pull down assays**. Cells were washed with ice-cold PBS and lysed in RIPA buffer, supplemented with protease inhibitor cocktail (Sigma) and phosphatase inhibitors (Roche), at 4 °C for 30 min. After 15 min centrifugation ($14,000 \times g$, 4 °C), the supernatant was pre-cleared by 1 h incubation with protein G Sepharose 4 Fast Flow (GE Healthcare Life Sciences). Cleared cell lysate was mixed with specific antibody overnight at 4 °C, then protein G Sepharose 4 Fast Flow was added and incubated for additional 3 h. Sepharose beads were pelleted by centrifugation, washed with cold lysis buffer, and then twice with cold PBS containing

1% Triton X-100 and 2 mM sodium orthovanadate. The pulled down material was eluted with 100 mM Glycine buffer, pH 3.0, and subjected to SDS-PAGE/WB assays with specific antibody. For ceramide measurements in the pulled down preparations, non-transfected (control) or A549 cells transfected with PCMV/CerS6 were washed with ice-cold PBS, suspended in hypotonic buffer (50 mM Tris-HCl, pH 7.4, 1 mM $MgCl_2$, with protease and phosphatase inhibitors, sonicated, and centrifuged 15 min at $14,000 \times g$. Supernatant was subjected to pull down with either p53-specific antibody or actin-specific antibody (control) as described above, with one exception: Triton X-100 was excluded from all of the buffers. Eluted material was analyzed for ceramide species levels by LC-MS/MS.

**LC-MS/MS analysis of sphingolipids**. Pulled down protein samples and cell lysates were stored at −80 °C prior to analysis. Lipid extraction and analyses were performed by the MUSC Lipidomics Shared Resource Analytic Core according to published procedure[53]. Briefly, samples fortified with internal standards were mixed with 2 ml of isopropyl alcohol:water:ethyl acetate (30:10:60 by volume), vortexed, and sonicated twice, followed by centrifugation (10 min at $4000 \times g$). The lipid extract (top layer, organic phase) was subjected to LC-MS/MS for sphingo-lipid analysis. Samples were normalized according to levels of respective proteins determined by immunoblot assays (pull down experiments) or to levels of lipid phosphate (denoted as $P_i$, experiments with cell lysates).

**Generation of truncated p53 proteins and p53 mutants**. Full-length human p53 coding sequence was sub-cloned into the pRSET B vector to produce a 6xHis-tagged protein. Site-directed mutagenesis with a Quick-Change Kit (Stratagene) was used to create constructs encoding for the His-tagged truncated p53 proteins: aa 1–300, 100–300 (DBD), 1–100. Single aa mutants of S240 and S241 were also obtained by site-directed mutagenesis of a full-length p53 construct. Oligonucleotide primers for site-directed mutagenesis are shown in Supplementary Table 1. Sequences of all constructs were confirmed by direct DNA sequencing.

**Expression and purification of p53**. *Escherichia coli* BL21(DE3) cells (purchased from Invitrogen) transformed with p53 expression constructs were grown on an LB/ampicillin plate overnight. A single colony was inoculated in 5 ml of LB medium containing 50 μg/ml ampicillin and grown overnight at 37 °C, the culture was transferred to 500 ml of LB medium with ampicillin, incubated at 37 °C until $OD_{600}$ reached 0.7 followed by addition of IPTG (1 mM final concentration, Thermo Fisher Scientific), and incubated additionally for 12 h at 25 °C. Cells were harvested ($5000 \times g$, 10 min), re-suspended in 50 ml of 50 mM $NaH_2PO_4$ buffer, pH 8.0 containing 300 mM NaCl, and incubated for 30 min with 1 mg/ml lysozyme at 4 °C. The suspension was chilled on ice, sonicated, and spun down at $10,000 \times g$ for 40 min. Supernatant was loaded on a PrepEase Ni–NTA high-yield agarose (USB Corporation) column and His-tagged protein was purified per the manufacturer's instructions. Purified protein preparations were further subjected to size-exclusion chromatography on Sephacryl S300 (GE Healthcare). The p53 construct containing 1–100 aa was subjected to an additional DE52 (Sigma Aldrich) anion exchange chromatography step.

**Modification of p53 DBD with pacFA ceramide**. The p53 100–300 construct was purified as described above and dialyzed into a buffer suitable for photoactivatable ceramide (PAC) modification (30 mM HEPES, 150 mM KCl, pH 8). The p53 protein was incubated with photoactivatable ceramide (50 μM) at room temperature for 1 h and the p53/PAC solution was run through a PD-10 desalting column (GE Helthcare) to remove any unbound photoactivatable ceramide (all steps were performed in the dark). The p53/PAC complex was transferred into the quartz cuvette and exposed to UV light for 20 min. As a control, p53 construct without photoactivatable ceramide was also exposed to UV light for 20 min. Confirmation of the modification was obtained by MALDI-TOF-MS using sinapinic acid as the matrix with data collected in linear positive ion mode, mass range 15,000–40,000 $m/z$. For MS calibration, a peptide calibration mixture 4700 (ABSciex) and BSA were used.

**Thermal shift assays**. Thermal shift assays were set up as 25 μl reactions in a real-time PCR plate (0.2 mg/ml p53, 50 mM Tris, 150 mM NaCl, 1 mM dithiothreitol (DTT), pH 7.6, 8× Applied Biosystems Protein Thermal Shift dye). The prepared plate was centrifuged to ensure that the solution was at the bottom of the wells and then covered with Microamp optical adhesive film. Fluorescence was measured using the QuantStudio 3 real-time PCR machine (Applied Biosystems) with x4m4 filters (excitation 580 ± 10 nm, emission 623 ± 14 nm) over temperature range of 25–90 °C.

**Proteomic analysis of ceramide-binding site**. Initially, unmodified control and PAC-modified p53 (100–300 aa residue construct) were precipitated by methanol/chloroform. Samples were denatured, reduced, and alkylated using 8 M urea in 100 mM Tris-HCl buffer, pH 7.8 containing 40 mM DTT and 40 mM iodoacetamide. The denatured, reduced, and alkylated samples were digested with endoproteinase AspN (Roche), 1:50 dilution, at 37 °C for 18 h. Trifluoroacetic acid (Sigma Aldrich) was added to the samples to 0.1%, and samples were purified and concentrated

using C18 ZipTip columns (Thermo Fisher Scientific) according to the manufacturer's protocol. Purified samples were dried and reconstituted in 60% acetonitrile for injection on the LTQ-Orbitrap XL mass spectrometer. On-line reversed-phase $C_{18}$ sample trapping, cleanup, and focusing was employed for the first 10 min of each analysis followed by 33-min elution gradient for analytical $C_{18}$ separation of the digested sample. The full scan profile MS at 60,000 resolution, 350–1800 $m/z$ was used for data collection. For fragmentation analysis, data collection used a top three in abundance selection for centroided tandem CID MS and a decision tree-based ETD activation fragmentation option. The .raw data files were analyzed by PEAKS Studio 7.5 proteomics database search engine to identify modifications present in peptides, using the following mass differences for: carbamidomethylation—57.02 Da, oxidation—15.99 Da, photoactivated ceramide—518.46 Da.

**Structural analysis**. To visualize possible binding arrangements, photoactivatable ceramide was docked to p53 (pdb ID: 2mej, residues 96–287) using Autodock VINA[54] with a $28 \times 28 \times 28 \text{ Å}^3$ target grid encompassing Ser 240–241. PyMol was used to visualize the p53 100–300 structure.

**Fluorescence quenching**. Steady-state fluorescence spectra (scanning range 260–340 nm) of purified recombinant p53 core domain (DBD, residues 100–300) were recorded on an F-7000 fluorescence spectrophotometer (Hitachi). The excitation wavelength was set at 270 nm with excitation and emission slit widths fixed at 5 nm. Fluorescence quenching was carried out with $PC_6$ or $PC_{16}$ (added as DMSO solution to a final concentration 10–2000 nM). Experiments were performed in triplicate using 1.0 μM protein in 10 mM Tris-HCl buffer, pH 7.4, containing 0.2 M NaCl and 10 mM β-mercaptoethanol. $K_d$ values were determined by non-linear fitting of the experimental data points (fluorescence emission at 304 nm) as described[55].

**Bimolecular fluorescence complementation (BiFC)**. For live-cell imaging of p53–MDM2 complexes, cells co-transfected with pCS2-P53-V1 and pCS2-MDM2-V2 plasmids (generous gift from Professor Cecilia Rodrigues) were grown on MatTek Chamber Slides (MatTek Corp) for 24 h. Fifty μM Nutlin 3 or 5 μM $PC_{16}$ were added 4 h after transfection. Images were captured using a Carl Zeiss LSM 710 laser scanning confocal microscope at the DHMRI and Cytell Cell Imaging System (GE Healthcare).

**siRNA**. Knockdown of CerS6 was performed using siRNA duplexes targeting sequence AACGCTGGTCCTTTGTCTTCA (Qiagen) as we previously described[11]. P53 was silenced using SignalSilence siRNA from Cell Signaling Technology. Scrambled siRNA with medium GC content (Invitrogen) was used as a control. Transfections were performed following the manufacturer's protocol.

**Cell cycle analysis**. Cell cycle analysis was carried out using propidium iodide staining/flow cytometry. Cells (about $1.0 \times 10^6$) were washed once with cold PBS and centrifuged at $100 \times g$ for 5 min. Pellets were resuspended in 200 μl of cold PBS, fixed by the addition of 4 ml of 70% ethanol, and incubated overnight at –20 °C. Following centrifugation, samples were resuspended in 500 μl of PBS containing 40 μg/ml propidium iodide and 100 μg/ml RNase (both from Roche, Basel, Switzerland), incubated for 30 min at 37 °C, and analyzed by fluorescence-activated cell sorter. Analysis was performed at the NORC Flow Cytometry Core, NRI UNC Chapel Hill, using (Becton Dickinson) or CytoFLEX flow cytometer (Beckman Coulter). Data analysis was performed using the FlowJo software (FlowJo, LLC).

**Sandwich ELISA assays**. Ninety-six-well plate was coated with capture antibody by incubating 100 μl of p53 polyclonal antibody solution (FL-393, 1 μg/ml, Santa Cruz) at 4 °C overnight. After the coating solution was removed and the plate was washed twice with PBST buffer (PBS containing 0.05% Tween 20), blocking was performed with 200 μl per well of 5% non-fat dry milk in PBS for 1 h at room temperature. The blocking solution was removed, the plate was washed twice in PBST, and 100 μl of pre-mixed recombinant p53/MDM2 solution (Boston Biochem), with or without $PC_{16}$ or Nutlin 3, were added per well. The plate was incubated for 4 h at room temperature, samples were removed, and wells were washed three times with PBST. For the detection of the p53–MDM2 complex, 100 μl of the MDM2 monoclonal antibody (ab137413, 1:5000 dilution, Abcam) was incubated at room temperature for 2 h, followed by triple wash with PBST and then by 2 h incubation with 100 μl of secondary sheep anti-mouse horseradish peroxidase-conjugated antibody (1:10,000, GE Healthcare Life Sciences, NA931). After removal of secondary antibody and washing the plate five times, 100 μl of tetramethylbenzidine substrate solution (Thermo Fisher Scientific) was added per well, incubated at room temperature for 30 min and reaction was stopped by adding 50 μl of Stop Solution (0.16 M sulfuric acid, Thermo Fisher Scientific). Absorbance was measured at 450 nm using VICTOR X5 multimode plate reader (PerkinElmer).

**p53 in vitro ubiquitination assays**. Reactions were prepared on ice in 0.5 ml polypropylene tubes using the MDM2/p53 Ubiquitination Kit (Boston Biochem)

according to the manufacturer's protocol. The reaction included 9 μl $H_2O$ and 3 μl each of the following (10× solutions): reaction buffer, p53 substrate protein, E1 enzyme, E2 enzyme, MDM2 enzyme, $Mg^{2+}$-ATP. Plates were gently mixed and the reaction was started by adding 3 μl 10× Ubiquitin solution. After incubation for 1 h at 37 °C, SDS-PAGE sample buffer containing 1.0 M DTT was added and samples were heat denatured at 98 °C for 5 min. Two control reactions (one with ATP omitted and another with 80 nM Nutlin 3) were run simultaneously. To determine the effect of $PC_{16}$ on the p53 ubiquitination, reactions were performed in the presence of increasing concentrations of ceramide (60, 80, and 100 nM). After denaturing, samples were analyzed by SDS-PAGE followed by WB using the manufacturer supplied p53 antibody.

## Data availability

Data supporting the findings of this manuscript are available from the corresponding author upon reasonable request. The mass spectrometry proteomics data have been deposited to the ProteomeXchange Consortium via the PRIDE[56] partner repository with the dataset identifier PXD010783.

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

## Acknowledgements

These studies were supported by the National Institute of Health Grants CA193782 (to N.I.K.); Z.M.S. and Lipidomics Shared Resource were supported by the National Institute of Health Grants P20 RR017677 (SC Lipidomics and Pathobiology COBRE) and 4P30GM103339-05 (COBRE in Lipidomics and Pathobiology, Core B). The authors thank Professor Cecilia Rodrigues for a generous gift of pCS2-P53-V1 and pCS2-MDM2-V2 plasmids; Professor Anthony Futerman for a generous gift of pcDNA3.1/CerS2-HA, pcDNA3.1/CerS3-HA, pcDNA3.1/CerS4-HA, pcDNA3.1/CerS5-HA, and pCMV2b/FLAG-CerS1 plasmids; and Dr. Yaroslav Tsybovsky for help with fitting the fluorescence quenching data and calculation of $K_d$ for PC$_{16}$.

## Author contributions

N.I.K. conceived the study and participated in the design of all experiments. S.A.K. participated in design of experiments, data interpretation, and in figure and manuscript preparation. B.F. designed and performed cell culture experiments, membrane-binding and pull down assays, and experiments involving fluorescence imaging. K.A.J. (equal contribution) performed purification of recombinant p53 100–300 construct, modified it with PAC and carried out thermal shift assays, proteolytic digestion, and worked with K.J.K. and D.R.K. on identification of the PAC-modified peptides. She also generated S240 and S241 p53 mutants and characterized their binding, recapitulated experiments with serum starvation and CerS1-6 expression in additional cell lines, and evaluated ceramide levels in these cells. A.E. generated p53 constructs, expressed and purified recombinant proteins, and performed fluorescence titrations. Z.M.S. synthesized derivatives of C$_{18-}$ C$_{16-}$ and C$_6$-ceramides. K.A.J. and D.A.H. performed modeling of the PAC into the p53 structure. All authors contributed to the discussion of results and participated in manuscript preparation.

## Additional information

**Competing interests:** The authors declare no competing interests.

