## [Peer Review File · Nature Communications]

Reviewers' Comments:

Reviewer #1:

Remarks to the Author:

the manuscript by Fekry et al describes the discovery of endogenous small molecule p53 activator - C16-ceramide. This study is novel, original and will open a new avenue in p53 research, a very interesting follow up studies are mentioned in the Discussion studies. The authors have applied a number of state-of-the-art molecular, biochemical and cellular approaches to demonstrate the direct binding of both endogenous and exogenous C16-ceramide to p53 in cells and in vitro. These experiments are very convincing. Moreover, the authors have mapped the region of interaction to the core domain of p53, namely, to residues S240,S241, and presented a model explaining how this interaction can prevent the binding of mdm2 to p53 and thus lead to p53 activation upon, for example, serum deprivation. Furthermore, the authors have shown that the prevention of C16-ceramide synthesis attenuates p53 accumulation and p53-mediated transcriptional response. I think this study is excellent and is worth of being published in Nature Communications.

There are a few minor comments to be addressed by the authors:

1. Fig2a: Control with avidin resin only - is missing

2. Fig.4:

on p. 13 it's written that A549 cells have been used for the experiments in Fig 4b, whereas Fig4b legend states that these were PC-3 cells

The order of panels d,e,f in Fig 4 do not correspond to the order of their description in the text

In Fig4e , left panel , arrow indicating p53 in mdm2 pull down should be removed, since p53 was not detected in this blot

3. Fig 6e: it would be good to indicate ceramide

4. Methods:

p.28: PAB240 is a monoclonal antibody, not polyclonal!

p.33: which polyclonal p53 antibody from Santa Cruz was used?

p.34: which Mdm2 antibody from Abcam was used?

Reviewer #2:

Remarks to the Author:

This is an interesting paper showing a novel mechanism for p53 activation. The data showing C16-ceramide binding to p53 is good, but less clear is the mechanism that then leads to stabilization and activation of p53. The study would be helped by further control experiments and an expansion of the data showing the importance of the interaction for the response to serum starvation and folate stress.

1. One general concern with the mechanisms is that the p53 in HeLa cells is degraded by E6/E6-AP. How does C16 ceramide work in these cells? A concern here is with the specificity of the observations.

2. The authors show that C16-ceramide will stabilise and activate p53 in A549, HepG2 and HeLa cells. They then show binding of C16 ceramide to p53 - interestingly C14 and C18 ceramide don't bind. The authors now have the reagents for perfect control experiments to support specificity of the results - cell permeable versions of C14 and C18 ceramide should not activate p53.

3. The analysis of binding site on p53 (Fig 2F) is very rudimentary – could this be refined further? It would be most informative to look at some of the most commonly occurring DNA binding domain point mutants. Does the binding depend on the conformation of p53? Do mutations of residues 240 or 241 affect binding? Would such p53 mutants (which are still subject to degradation by MDM2) be resistant to the effects of C16 ceramide?

4. The pull down shown in Fig 2A isn't convincing – it needs a control. C14 and C18 ceramide and a non-binding p53 point mutant could help here.

5. The inhibition of the p53/MDM2 interaction is interesting, given the main MDM2 binding site on p53 is at the N-terminus of the protein (this is the site blocked by Nutlin, used here as a positive control). While some interaction of MDM2 with different regions of p53 has been reported, most mutations in the DNA binding domain of p53 do not prevent MDM2 binding. How is this inhibition of MDM2 binding working? And (see above) is the binding to E6/E6AP also affected?

6. Does C16 ceramide affect p53/DNA binding? Figure 6E suggests that ceramide binding is lost as the p53 enters the nucleus, but I see no evidence for this. If the ceramide binding persists, does this affect p53-dependent transcription?

7. Both p53 and Mdm2 are nuclear protein, and the targeting of p53 for degradation is likely to occur in the nucleus. This seems at odds with the model presented by the authors. What is the evidence that CerS6 expression leads to the nuclear translocation, rather than simply accumulation, of p53?

8. The binding of p53 to C16 ceramide following expression of CerS6 is clear (Fig5A and B) but any interpretation of the lack of effect of CerS1-5 would need a demonstration that these enzymes are increasing the levels of other ceramide species as expected (as shown for CerS6 in Fig 5A).

9. The final experiment shows some evidence for the physiological relevance of the observations in response to serum starvation. The authors show that serum starvation leads to elevated levels of C16, a response that is abrogated by siRNA depletion of CerS6. P53 is nicely shown to be in complex with C16 ceramide, but not other ceramide species (Figure 6C and D). This should be shown in all 3 cell lines used for the earlier experiment (it's not clear why the authors switch to HCT116 cells here). The effect on cell cycle should also be shown in the other cell lines along with some indication of the reproducibility of this result. What happens to other p53 target genes?

10. A similar response to folate stress is shown in the supplementary data – I think this is important and these data should be expanded to match those shown for serum starvation.

Reviewer #3:

Remarks to the Author:

This is an interesting and important study identifying a novel interaction between the lipid C16 ceramide and the p53 tumour suppressor. P53 occupies a central position in cancer research, so the work is of broad significance. In a sense this work is incremental on the authors' previous work, which has already established a reciprocal relationship between C16 ceramide, synthesized by CerS6, p53, and folate stress or chemotherapy-induced stress. The major advance in this paper is the demonstration of a direct interaction between C16 ceramide and p53, using multiple different biochemical approaches, and this is important. One point that is not addressed in this study is how C16 ceramide, a very hydrophobic lipid formed at the endoplasmic reticulum, would access to p53.

Although the topic is of interest, there are a number of poorly controlled or poorly replicated experiments, and the figures are often lacking detail. In many instances the labelling is too small and not legible. Figure 4e is essentially uninterpretable in its current format and needs to be much more clearly presented to enable the reader to understand what is presented in each blot, with appropriate controls (a non-specific antibody control, as used in Figure 6?).

- Figure 2a requires an appropriate control, e.g. biotin alone or a different ceramide-biotin conjugate.

- Some of the results, particularly 4d, 5c, 5e, and 6b, should be shown for more than one cell line. In fact, it is often unclear which cells are being used for which experiment, and this needs to be made clearer in the figure legends. Ideally, the authors would show at least one non-cancer cell line as well. On this point, the text needs to clarify p53 WT vs p53 mutant cell lines. I don't believe any experiments have been done with p53 null lines, but this isn't clear. What would happen with CerS6 up-regulation, ceramide levels, cell cycle arrest, and sensitivity to the C16 ceramide analogue in cells lacking functional p53 and exposed to serum deprivation?
- A single experiment with 4 replicates is not sufficient for Figure 4d. This figure shows viability, not proliferation, as stated in the text.
- In Figure 6b, a control for cell viability/morphology is required, i.e. not just the Venus fluorescence. Is loss of fluorescence related to loss of viability? Note text refers to A549 cells but legend refers to PC-3 cells.
- Another important point that I feel should be addressed is that selectivity for binding to C16 ceramide has been demonstrated in a number of assays, yet the C16 ceramide pyridinium analogue, which differs in structure more than C16 vs C18 ceramide, is apparently binding as well. So does the C18 ceramide pyridinium analogue that has previously been described (Sentelle, Nat Chem Biol, 2012) not bind to p53? Do those two extra carbons exclude the molecule from the binding site? It would also be good if the authors can carry out the experiment in Figure 2b with natural ceramide.
- Would C16 pyridinium ceramide be expected to localise to the mitochondria (Sentelle et al, 2012)?
- Statistical tests are not adequately described throughout. What statistical tests were used for comparing lipid levels in figures 5 and 6? What are the repeat measures in Fig. 2e?
- The experiments using LC-MS/MS to show that C16 ceramide is enriched in p53 immunoprecipitates are important, as they demonstrate the specific interaction with C16 ceramide in living cells. However, CerS5 is also localised to the ER and produces C16 ceramide. Why would CerS5 overexpression not stabilise p53? The authors discuss this point but only briefly.
- The Introduction and/or Discussion needs to include a paragraph dealing with the background on ceramide and p53. One sentence in the discussion is not enough.

Reviewer #4:

Remarks to the Author:

Accept as is

We appreciate the detailed comments from the reviewers, particularly their recognition of the importance and novelty of the ceramide-p53 interaction. Predominant issues noted by the reviewers included several missing control experiments, use of different cell lines for different experiments, and insufficient discussion of biological relevance. In response, we have addressed/discussed all reviewer's concerns, most of them with additional experiments, which provided new data. We have also addressed all technical issues, such as panel labeling, antibody used, statistical analysis, etc. Our responses to specific reviewer comments are as follows:

Reviewer 1

1. Fig. 2a: Control with avidin resin only - is missing

Data from the control experiment (purified p53 or A549 cell lysate were incubated with biotin and then pulled-down on CaptAvidin column) are now included in Fig.2a (Con).

2. Fig.4:

on p. 13 it's written that A549 cells have been used for the experiments in Fig 4b, whereas Fig4b legend states that these were PC-3 cells

PC-3 cells (p53-null) were used in the BIFC experiments to prevent oligomerization of V1-p53 or p53-V2 with endogenous cellular p53, that would decrease the formation of complexes producing fluorescence signal. We have corrected the text accordingly.

The order of panels d,e,f in Fig. 4 do not correspond to the order of their description in the text

We have rearranged and re-labeled the panels in Fig. 4 to match the order of citation in the text.

In Fig. 4e, left panel, arrow indicating p53 in mdm2 pull down should be removed, since p53 was not detected in this blot

For the Fig. 4e, we have removed the p53-labeled arrows and replaced them with 55 KDa molecular weight markers.

3. Fig 6e: it would be good to indicate ceramide

The legend for C₁₆-ceramide has been included in the schematic, Fig.6e.

4. Methods:

p.28: PAB240 is a monoclonal antibody, not polyclonal!

p.33: which polyclonal p53 antibody from Santa Cruz was used?

p.34: which Mdm2 antibody from Abcam was used?

PAb240 description is corrected in methods section (p. 29).

The antibody information has now been listed (p. 35).

Reviewer 2

1. One general concern with the mechanisms is that the p53 in HeLa cells is degraded by E6/E6-AP. How does C16 ceramide work in these cells? A concern here is with the specificity of the observations.

We were also puzzled to see the p53 response to ceramide in HeLa cells. However, when we examined relevant literature, we realized that the mechanism in these cells could be quite similar to the mechanism proposed for other cell lines. The function of E6/E6-AP with regard to p53 in HeLa cells is similar to the function of MDM2 in most cell lines. Specifically, **E6 binds p53 in the proximity to the ceramide binding region**. Then E6 recruits an E3 ligase to ubiquitinate p53 and target it for degradation. Thus, the ceramide protection of p53 from degradation in this case would be mechanistically the same. We have discussed such mechanism in the original manuscript, but we regret that we did not re-iterate on this issue. Discussion of this matter in the revised manuscript is presented on pp. 24-25.

2. The authors show that C16-ceramide will stabilize and activate p53 in A549, HepG2 and HeLa cells. They then show binding of C16 ceramide to p53 - interestingly C14 and C18 ceramide don't bind. The authors now have the reagents for perfect control experiments to support specificity of the results - cell permeable versions of C14 and C18 ceramide should not activate p53.

Unfortunately, none of the pyridinium ceramides except PC₆ are commercially available. We have been able to obtain C₁₈-pyridinium ceramide reported previously (Sentelle, et al, Nat Chem Biol 2012) and demonstrated that similar to natural C₁₈-ceramide this derivative does not bind to p53 (new Supplementary Fig. 2). We had only a very limited amount of this compound, not sufficient to perform extensive studies.

3. The analysis of binding site on p53 (Fig. 2F) is very rudimentary – could this be refined further? It would be most informative to look at some of the most commonly occurring DNA binding domain point mutants. Does the binding depend on the conformation of p53? Do mutations of residues 240 or 241 affect binding? Would such p53 mutants (which are still subject to degradation by MDM2) be resistant to the effects of C16 ceramide?

For the revised manuscript, we have generated several mutants of p53 (residues 240 and 241 were replaced) and demonstrated that substitutions at either of these positions can completely eliminate the C16-ceramide binding (new Supplementary Fig. 4 and pp. 12-13 and 23 of revised manuscript), proving the role of these residues as well as the specificity of the ceramide binding. The ligand docking routines we used are unable to identify a unique position of the ceramide alkyl side chains and the only experimental contact was determined with the PAC experiment. We thus would not be able to experimentally verify a more detailed computational binding site analysis. The DNA binding domain mutant R175H (one of the most frequent mutations in cancer cells) was also examined in these experiments and was shown to bind C₁₆-ceramide similar to the

WT protein (Supplementary Fig. 4 and p. 13). Apparently, at least some common p53 mutants will be protected by ceramide similar to the WT protein.

4. The pull down shown in Fig 2A isn't convincing – it needs a control. C14 and C18 ceramide and a non-binding p53 point mutant could help here.

As a control for the pull-down experiments, incubation of p53 or cell-lysate with biotin was used (instead of the biotinylated-C₁₆-ceramide). As expected, when incubated with biotin itself p53 was not pulled down. These data are now included in Fig. 2a, labeled as Con. Taking into consideration that biotinylated ceramides are not commercially available, we have chosen a different approach to investigate the p53 selectivity. We believe that our data demonstrating the pulldown of p53 in the complex with natural endogenous C16-ceramide, but not other ceramide species (Figs. 5&6 and Supplementary Figs. 10&11) support the discovered interaction better.

5. The inhibition of the p53/MDM2 interaction is interesting, given the main MDM2 binding site on p53 is at the N-terminus of the protein (this is the site blocked by Nutlin, used here as a positive control). While some interaction of MDM2 with different regions of p53 has been reported, most mutations in the DNA binding domain of p53 do not prevent MDM2 binding. How is this inhibition of MDM2 binding working? And (see above) is the binding to E6/E6AP also affected?

Though less appreciated than the interaction with the p53 N-terminal domain, the interaction of MDM2 with the p53 DBD has been studied functionally and structurally (Yu, et al, PNAS, 2006). Importantly, the central region of MDM2 interacting with the DBD is believed to be involved in the ubiquitination and degradation of p53. To clarify the issue of how ceramide could interfere with the p53-MDM2 interaction, we have mapped the MDM2-binding interface of the p53 in our model with bound ceramide (new Fig. 3d, the interface is shown in *yellow*). This model shows that ceramide is accommodated in the middle of the interface between two surface regions making contacts with MDM2. Corresponding discussion (p. 23) explains how we envision the ceramide interference with the p53-MDM2 complex. This discussion also covers the mechanism of the p53 rescue by ceramide in HeLa cells (also discussed above in our response to comment #1).

6. Does C16 ceramide affect p53/DNA binding? Figure 6E suggests that ceramide binding is lost as the p53 enters the nucleus, but I see no evidence for this. If the ceramide binding persists, does this affect p53-dependent transcription?

We did not investigate a question of whether the ceramide is still bound to the protein as p53 enters the nucleus. It should be noted, however, that activation of the downstream targets of p53 such as p21 (under serum deprivation, PC₁₆-treatment or folate stress) or Puma (in case of folate stress) have been demonstrated (Fig. 1b). This means that if ceramide binding to p53 persists in the nucleus, it does not affect the activation of at least a subset of target genes.

7. Both p53 and Mdm2 are nuclear proteins, and the targeting of p53 for degradation is likely to occur in the nucleus. This seems at odds with the model presented by the

authors. What is the evidence that CerS6 expression leads to the nuclear translocation, rather than simply accumulation, of p53?

Numerous studies demonstrated that both p53 and MDM2 can be localized to cytoplasm and are constantly moving between cytosol and the nucleus (see for example Green, Kroemer, Nature, 2009; O'Brate, Giannakakou, Drug Resistance Updates, 2003). As well, the degradation of p53 can take place in both compartments (Yu, Geyer, Maki, Oncogene, 2000). Overall, healthy unstressed cells maintain low levels of cytoplasmic p53 due to the negative regulation by MDM2 (Hock, Vousden, BBA, 2014). A canonical view of the p53 signaling mechanism described elsewhere includes; the release of p53 from MDM2-driven ubiquitination in cytoplasm (this can be enable by myriads of modifications), accumulation of p53 in cytoplasm due to the loss of ubiquitin-dependent degradation, the translocation of accumulated p53 into the nuclei, where it functions as a transcription factor. We demonstrated such mechanism in our previous studies (Oleinik et al, Biochem J, 2005). As well, such time-dependent shift in the p53 localization can be seen in Figs. 1c&5c.

8. The binding of p53 to C16 ceramide following expression of CerS6 is clear (Fig. 5a&b) but any interpretation of the lack of effect of CerS1-5 would need a demonstration that these enzymes are increasing the levels of other ceramide species as expected (as shown for CerS6 in Fig 5a).

Corresponding experiments have been performed. New Supplementary Fig. 9 shows enzyme-specific changes in ceramide levels upon transfection of A549 cells with individual ceramide synthases. These new data address Reviewer's question and are discussed in pp. 17-18 of the revised manuscript.

9. The final experiment shows some evidence for the physiological relevance of the observations in response to serum starvation. The authors show that serum starvation leads to elevated levels of C16, a response that is abrogated by siRNA depletion of CerS6. P53 is nicely shown to be in complex with C16 ceramide, but not other ceramide species (Figure 6C and D). This should be shown in all 3 cell lines used for the earlier experiment (it's not clear why the authors switch to HCT116 cells here). The effect on cell cycle should also be shown in the other cell lines along with some indication of the reproducibility of this result. What happens to other p53 target genes?

As suggested, new experiments with serum deprivation have been performed in HCT116 and HepG2 cell lines including; (i) the cell cycle evaluation, (ii) ceramide measurements and (iii) p53 pull-downs with specific antibody and subsequent measurements of bound ceramides, as well as (iv) demonstration that CerS6 siRNA knockdown rescues serum starvation responses. Essentially, these experiments recapitulate all effects observed in A549 cells and are included as new Supplementary Figs. 10 and 11 of the revised manuscript. Additional experiments demonstrated that absence of p53 partially protects HCT116 and HepG2 cells from PC₁₆ toxicity (new Supplementary Fig. 8). We agree with the reviewer that it would be interesting to evaluate the response of additional p53 target genes; this is a goal for future investigation.

10. A similar response to folate stress is shown in the supplementary data – I think this is important and these data should be expanded to match those shown for serum starvation.

We completely agree with Reviewer regarding connection of folate stress with ceramide generation and signaling. Such studies (cell cycle, ceramide measurements, downstream targets activation, etc.) have been completed and published (Hoeflerlin, et al, JBC, 2013). Actually, these folate studies (referenced in the manuscript) have instigated our interest to the mechanisms of ceramide effects.

Reviewer 3

1. One point that is not addressed in this study is how C16 ceramide, a very hydrophobic lipid formed at the endoplasmic reticulum, would access to p53.

We agree with Reviewer that the question of accessing p53 by a hydrophobic ceramide is very interesting and important. We acknowledge this and several other important questions in the Discussion section (p. 25). Since p53 is not the only non-membrane protein capable of ceramide binding (CERT, I2PP2A), mechanisms to load these hydrophobic molecules on such proteins do exist in the cell, and we are at present performing experiments to investigate them as applicable to p53. We believe that the data presented in our manuscript convincingly demonstrate highly specific binding of C₁₆-ceramide to p53, both *in vitro* and *in vivo*, that prevents p53 interaction with MDM2 and leads to p53 up-regulation and activation. Our data also demonstrate the biological relevance of the discovered mechanism of the p53 regulation by ceramide.

2. Although the topic is of interest, there are a number of poorly controlled or poorly replicated experiments, and the figures are often lacking detail. In many instances the labelling is too small and not legible. Figure 4e is essentially uninterpretable in its current format and needs to be much more clearly presented to enable the reader to understand what is presented in each blot, with appropriate controls (a non-specific antibody control, as used in Figure 6?).

We regret the poor annotation and unclear labeling in some figures. We have re-done Fig. 4e and changed the labelling with a hope that this improves the data presentation. More details regarding the cell lines and statistical analysis have been added to figure legends. Additionally, we have redone many of the experiments and documented the controls (noted in responses to other reviewers).

3. Figure 2a requires an appropriate control, e.g. biotin alone or a different ceramide-biotin conjugate.

Control with biotin has been added to Fig. 2a as suggested. As expected, when biotin was used instead of biotinylated C₁₆-ceramide, p53 was not pulled down. Biotinylated C₁₆-ceramide was synthesized by one of the co-authors and other such derivatives are not available, so additional controls were not performed. However, our data demonstrating the pulldown of p53 in the complex with natural endogenous C₁₆-ceramide, but not other ceramides present in the cells at

higher concentrations (Figs. 5&6 and Supplementary Figs. 10&11), support the specificity of discovered interactions much better.

4. Some of the results, particularly 4d, 5c, 5e, and 6b, should be shown for more than one cell line. In fact, it is often unclear which cells are being used for which experiment, and this needs to be made clearer in the figure legends. Ideally, the authors would show at least one non-cancer cell line as well. On this point, the text needs to clarify p53 WT vs p53 mutant cell lines. I don't believe any experiments have been done with p53 null lines, but this isn't clear. What would happen with CerS6 up-regulation, ceramide levels, cell cycle arrest, and sensitivity to the C16 ceramide analogue in cells lacking functional p53 and exposed to serum deprivation?

We regret that we did not explicitly state the p53 status of our cell lines in the original manuscript. We have clarified this issue in the revised manuscript. Overall, the following p53-positive cell lines were used, all with the WT p53: A549, HCT116, and HepG2. As suggested by Reviewer, we have performed additional experiments using matching cell lines lacking functional p53. Specifically, A549 p53^{-/-} and HepG2 p53^{-/-} cells were generated by the p53 siRNA silencing (described in the manuscript); HCT116 p53^{-/-} cell line is isogenic to HCT116 (p53^{+/+}) (generous gift from Dr. Vogelstein). Initial experiments also included p53-positive HeLa cells, also WT for p53. PC-3 cell line was selected for BIFC experiments since it is p53-null (the null p53 status prevents the oligomerization of V1-p53 or p53-V2 with endogenous cellular p53 that does not have Venus tags). New data for p53^{-/-} cells are presented in Fig. 4f and in Supplementary Figs. 5,6&8.

5. A single experiment with 4 replicates is not sufficient for Figure 4d. This figure shows viability, not proliferation, as stated in the text.

Experiment in question has been repeated three times, each in quadruplicate. The figure (Fig. 4f of the revised manuscript) shows a representative experiment. We have made changes in the text of the manuscript and in the figure legend to clarify this issue. Of note, for the revised manuscript similar experiments using additional cell lines have been added (new Supplementary Fig. 8). All these experiments show essentially the same outcome, much weaker response to PC₁₆ upon the p53 loss.

6. In Figure 6b, a control for cell viability/morphology is required, i.e. not just the Venus fluorescence. Is loss of fluorescence related to loss of viability? Note text refers to A549 cells but legend refers to PC-3 cells.

Reviewer perhaps refers here to Fig. 4b (Venus fluorescence). We have added the panels with overlay of the fluorescence- and phase-contrast images as new Supplementary Figs. 5&6. It can be easily ascertained from these images that the samples treated with either Nutlin-3 or PC₁₆ do not differ by morphology or by cell numbers from non-treated control samples. We also added a statement in the manuscript indicating that cells were treated with drugs for a short period of time (18 h) which is not sufficient to induce noticeable antiproliferative effects. Corrections in

the text have been made to indicate the PC-3 cells as the line used for these experiments, as well as explain a rationale for such selection.

7. Another important point that I feel should be addressed is that selectivity for binding to C16 ceramide has been demonstrated in a number of assays, yet the C16 ceramide pyridinium analogue, which differs in structure more than C16 vs C18 ceramide, is apparently binding as well. So does the C18 ceramide pyridinium analogue that has previously been described (Sentelle, Nat Chem Biol, 2012) not bind to p53? Do those two extra carbons exclude the molecule from the binding site? It would also be good if the authors can carry out the experiment in Figure 2b with natural ceramide.

In Fig. 2, ceramides indicated as C₁₂ to C₂₄ in panel c are natural ceramides, while PC₆ and PC₁₆ are pyridinium derivatives of the natural ceramides with the pyridinium group on the acyl chain. Correspondingly, dhC₁₆ is a natural dihydro-C₁₆-ceramide. We have added this information in the text and figure legends to make it more clear. Neither C₁₄, nor C₁₈ pyridinium ceramides with the pyridinium on the acyl chain are currently available. However, we have additionally tested the C₁₈-pyridinium ceramide (LCL461, referenced by Reviewer) with the pyridinium group on the sphingoid base and demonstrated that this ceramide, unlike to PC₁₆, does not interact with p53 in the membrane binding assay (new Supplementary Fig. 2). Our data on both natural ceramide binding and on the analysis of the ceramides that were pulled-down from stressed cells by p53 Ab (bound to the p53 protein), indeed, demonstrate, that the difference in two carbon atoms is recognized by p53. While the molecular basis for this recognition is not clear at present, such high specificity of binding by p53 is not unique. The work referenced by Reviewer demonstrates that the PP2A inhibitor discriminates between C₁₈- and C₁₆-ceramides as well. We have extended the discussion of this matter in the revised manuscript (p. 23). We completely agree with Reviewer that fluorescence titration with natural ceramides would be very beneficial. Unfortunately, these experiments are not feasible because long-chain ceramides are extremely hydrophobic and form condensed films (Szulc, et al, Bioorganic and Med. Chemistry, 2006), not micelles in aqueous solutions. In agreement with these findings, our attempts to use natural ceramide in the titrations failed.

- 8.** Would C16 pyridinium ceramide be expected to localize to the mitochondria (Sentelle et al, 2012)

Indeed, mitochondrial accumulation of pyridinium ceramide derivatives including PC16 (Dindo et al., 2006) has been reported. Though in our study we did not specifically focus on the subcellular localization of PC₁₆, the above referenced paper indicated that PC₁₆ is accumulated in extramitochondrial compartments as well (with 80% of the total PC16 being outside of mitochondria), that would explain the p53-associated effect of this compound.

9. Statistical tests are not adequately described throughout. What statistical tests were used for comparing lipid levels in figures 5 and 6? What are the repeat measures in Fig. 2e?

Student's t-test has been used for statistical analysis of the differences. We have added the information on statistical analysis to all figure legends, as well as the repeat measures to figure 2e (three independent experiments performed, each in triplicate).

10. The experiments using LC-MS/MS to show that C16 ceramide is enriched in p53 immunoprecipitates are important, as they demonstrate the specific interaction with C16 ceramide in living cells. However, CerS5 is also localized to the ER and produces C16 ceramide. Why would CerS5 overexpression not stabilize p53? The authors discuss this point but only briefly.

Our new data (new Supplementary Fig. 9) and data in Fig. 5 demonstrate, in several cell lines, that CerS5 overexpression did not activate p53, even though this enzyme also produces C₁₆-ceramide. It is not completely clear at present, why CerS5-produced ceramide did not interact with p53. The most likely explanation is that CerS6 produces more C16-ceramide than CerS5 (new Supplementary Fig. 9) exceeding the required concentration threshold. Several other hypothetical mechanisms could be proposed for the disparity of the fate of ceramides produced by different synthases, including the enzyme localization/membrane orientation, different interacting partners (both in the membrane and in the cytosol) and the ability to release the ligand to a specific target. We have added more discussion of this issue in the revised manuscript (pp. 24 and 25).

11. The Introduction and/or Discussion needs to include a paragraph dealing with the background on ceramide and p53. One sentence in the discussion is not enough.

Unfortunately, available information on the regulation of p53 by ceramide is scarce and to some extent inconsistent, and we emphasize this point in our manuscript. We have also added more discussion of this matter as suggested (p. 21).

Reviewer 4.

We are grateful to the Reviewer for accepting our work.